# Monitoring the Foliar Nutrients Status of Mango Using Spectroscopy-Based Spectral Indices and PLSR-Combined Machine Learning Models

Gopal Ramdas Mahajan [1,*], Bappa Das [1], Dayesh Murgaokar [1], Ittai Herrmann [2], Katja Berger [3], Rabi N. Sahoo [4], Kiran Patel [1], Ashwini Desai [1], Shaiesh Morajkar [1] and Rahul M. Kulkarni [1]

1 Natural Resource Management, ICAR–Central Coastal Agricultural Research Institute, Goa Velha 403402, India; bappa.das@icar.gov.in (B.D.); thenatureboy205@gmail.com (D.M.); kiranpatel091@gmail.com (K.P.); ashwinik1993@yahoo.com (A.D.); shaiesh119211@gmail.com (S.M.); rahul.kulkarni@icar.gov.in (R.M.K.)
2 The Plant Sensing Laboratory, The Robert H. Smith Institute for Plant Sciences and Genetics in Agriculture, The Robert H. Smith Faculty of Agriculture, Food and Environment, The Hebrew University of Jerusalem, P.O. Box 12, Rehovot 7610001, Israel; ittai.herrmann@mail.huji.ac.il
3 Department of Geography & Remote Sensing Ludwig-Maximilians-Universität München, 80333 Munich, Germany; katja.berger@lmu.de
4 Division of Agricultural Physics, ICAR–Indian Agricultural Research Institute, New Delhi 110012, India; rabi.sahoo@icar.gov.in
* Correspondence: gopal.mahajan@icar.gov.in; Tel.: +91-0832-2284679

**Abstract:** Conventional methods of plant nutrient estimation for nutrient management need a huge number of leaf or tissue samples and extensive chemical analysis, which is time-consuming and expensive. Remote sensing is a viable tool to estimate the plant's nutritional status to determine the appropriate amounts of fertilizer inputs. The aim of the study was to use remote sensing to characterize the foliar nutrient status of mango through the development of spectral indices, multivariate analysis, chemometrics, and machine learning modeling of the spectral data. A spectral database within the 350–1050 nm wavelength range of the leaf samples and leaf nutrients were analyzed for the development of spectral indices and multivariate model development. The normalized difference and ratio spectral indices and multivariate models–partial least square regression (PLSR), principal component regression, and support vector regression (SVR) were ineffective in predicting any of the leaf nutrients. An approach of using PLSR-combined machine learning models was found to be the best to predict most of the nutrients. Based on the independent validation performance and summed ranks, the best performing models were cubist ($R^2 \geq 0.91$, the ratio of performance to deviation (RPD) $\geq 3.3$, and the ratio of performance to interquartile distance (RPIQ) $\geq 3.71$) for nitrogen, phosphorus, potassium, and zinc, SVR ($R^2 \geq 0.88$, RPD $\geq 2.73$, RPIQ $\geq 3.31$) for calcium, iron, copper, boron, and elastic net ($R^2 \geq 0.95$, RPD $\geq 4.47$, RPIQ $\geq 6.11$) for magnesium and sulfur. The results of the study revealed the potential of using hyperspectral remote sensing data for non-destructive estimation of mango leaf macro- and micro-nutrients. The developed approach is suggested to be employed within operational retrieval workflows for precision management of mango orchard nutrients.

**Keywords:** chemometrics; hyperspectral remote sensing; multivariate modeling; precision nutrient management; VNIR spectroscopy

## 1. Introduction

Over the last two decades, advancements in remote sensing technologies such as the use of reflectance spectroscopy, airborne and satellite technology, and statistical analysis approaches thereof have made it easy to understand several key processes and components of plants such as plant population [1–3], grain yield and biomass [4–8], pigment or chlorophyll [9–11], water stress response [12–15], nutritional status [16–21] or pest and disease identification [22–25]. Yet, in-field proximal sensing to estimate the nutritional status of the

crops is an economical and technical challenge [26]. The commonly employed statistical approaches to retrieve the information on plant biomass, pigment, and foliar nutrient status from hyperspectral data include (1) linear regression methods—reflectance, derivative types, reflectance transformation, narrowband vegetation indices (VIs), (stepwise) multiple linear regression, principal component analysis/regression (PCA/PCR), partial least square regression (PLSR), (2) non-linear regression methods—random forest regression (RFR), artificial neural network (ANN), support vector regression (SVR), Gaussian process regression (GPR), (3) physically-based methods—radiative transfer model (RTM) inversion, and (4) Hybrid methods—combination of minimum two methods [27,28], usually machine learning regression model which is trained over a RTM data base. The VIs are established to enhance sensitivity to vegetation characteristics [29], while minimizing confounding factors such as soil background, canopy geometry, leaf optical properties, and atmospheric conditions [30,31]. Previously, VIs have been extensively used for estimating and mapping foliar nitrogen (N) [28], water and chlorophyll contents [28,32–38], phosphorus (P), sulfur (S), and potassium (K) [16,39,40]. In [41], the oil palm nutrient content was retrieved using VIs. A normalized difference and simple ratio of the reflectance at 1423 nm and 1877 nm correlated with N with $R^2 = 0.53$, while that of 1164 nm and 1238 nm correlated to calcium (Ca) with $R^2 = 0.50$. The P, K, magnesium (Mg), boron (B), copper (Cu), and zinc (Zn) had moderately positive correlations ($R^2$ of 0.33–0.49). The results obtained by them were better than those for the commonly used or published VIs. A canopy reflectance-based ratio of difference index was used successfully by [42] to estimate the leaf N content at different growth stages of litchi and achieved accuracies with a coefficient of determination ($R^2$) > 0.50 and root means square error (RMSE) < 0.14. The most frequently used hyperspectral VIs are simple ratio and normalized difference index derived from two or more narrow wavelengths or wavebands. The advantage of VI is that they are simpler and use a few sensitive wavelengths and wavebands, however, the disadvantage is that the information available in other parts of the spectrum is not considered or lost [43].

At the field level, predicting foliar or canopy mineral nutrients other than N are less effective particularly for P, K, and S [44], and other macro- and micronutrients using conventional statistical analysis of the spectral data. However, rapid development in spectroscopy for plant mineral nutrient analysis has opened up for use of visible (VIs, 400–700 nm) and near infra-red (NIR, 700–1300 nm) range for nutrients such as K [45] and Ca and Mg [46]. Modeling with the spectral reflectance data provides a convenient and interpretable means to understand the fundamental interactions of plant conditions with radiant energy detected by multispectral or hyperspectral sensors. This comparison might help to understand their comprehensive performance from the perspective of model accuracy, simplicity, robustness, and interpretation and would guide in selecting the optimal method for estimating plant nutrients at various levels. Modeling of the hyperspectral imaging and non-imaging spectroscopic data has been proven to capture the variations in biochemical components in plant leaves and canopies such as chlorophyll, N, P, and K [30,32,47]. Multivariate analysis models (i.e., chemometrics) such as multiple linear regression (MLR), stepwise MLR, partial least square regression (PLSR), principal component regression (PCR), etc. have been used to characterize the plant nutrient status from the spectral characteristics. A reasonable apple leaf N prediction accuracy was achieved by [48] using PLSR and MLR of raw reflectance ($R^2 = 0.77$ and 0.78, respectively) and first-derivative reflectance ($R^2 = 0.77$ and 0.77, respectively). [49] achieved significant prediction of the Mg, P, S, K, and Ca of the tallgrass prairie vegetation using PLSR of the normalized difference standardized data in the wavelength range of 470–800 nm. The spectral data standardization by normalized difference reduced the background interference in the leaf reflectance. A multivariate analysis with PCR of spectral data could predict the macro- and micro-nutrients in the oil palm leaves with $R^2 = 0.56$–0.90 [41]. [50] successfully selected rubber tree leaf P sensitive wavelengths using the Monte Carlo-uninformative variable elimination combined with the successive projections algorithm and predicted it using MLR with a prediction accuracy of $R^2 = 0.69$. Using PLSR of the hyperspectral image

data in a wavelength range of 500–1700 nm of maize and soybean, [51] determined the macronutrients content (N predicted best followed by P, K, and S) ($R^2$ = 0.69–0.92 and ratio of performance to deviation (RPD) = 1.62–3.62) and micronutrients (Cu and Zn were best predicted, followed by Fe and Mn) ($R^2$ = 0.19–0.86, RPD = 1.09–2.69) satisfactorily. The predictions of sodium and B were not satisfactory. Machine learning models are a unique and robust technique to analyze and model any data being non-linear and non-parametric [21,52,53]. Use of a combination of linear multivariate analysis models such as PLSR and principal component analysis (PCA) with non-linear and non-parametric models such as artificial neural network (ANN) [54–57] elastic net (ELNET), support vector regression (SVR), Gaussian process regression (GPR), multivariate adaptive regression spline (MARS), random forest (RF), extreme gradient boosting (XGB), generalized additive model (GAM), and k-nearest neighbor (KNN) [20] has been reported to retrieve information from spectral features. The machine learning models are capable of performing numerous calculations in several combinations and are thus useful to reduce the time involved in the analysis. In [21], the authors predicted the citrus, Valencia-orange, leaf N, P, K, Ca, Mg, S, Cu, iron (Fe), manganese (Mn), and Zn by using RF, ANN, and KNN of the spectral reflectance and its first-order derivative with $R^2$ = 0.61 to 0.91. An RF and support vector machine regression (SVMR) of the airborne hyperspectral remote sensing imagery data were used by [58] to estimate the N, P, K, Zn, Na, Cu, and Mg with $R^2$ = 0.55–0.78 (with RF) and S and Mn with $R^2$ = 0.68–86 (with SVMR) of the mixed pasture in New Zealand. They emphasized a better performance of the non-linear machine learning model (SVMR and RF) than the linear (PLSR). Some of the commonly employed multivariate modeling techniques to extract information from hyperspectral data and to establish a relationship between spectral reflectance and measured variables are stepwise MLR [59,60], PLSR [61,62], successive projections algorithm coupled with MLR [63,64], ANN [63,65], and SVMR [47,66]. Very recently, a combination of the PCA and PLSR combined learning models have been used by [20] as a non-destructive tool, to predict the leaf ion (K, Na, Ca, and Mg) content for phenotyping of rice to salt-stress. They found the prediction accuracy of different approaches in order as PLSR-combined models > PCA-combined models > indices-based models. A non-linear SVR based radial basis function (RBF) kernel predicted critical N concentration in the sugarcane canopy correlated with actual N by $R^2$ of 0.78 and RMSE of 0.035% [67]. Nutrients with low plant or leaf concentration and subtle physical absorption features still pose a challenge and less attention has been paid for its error-free estimation using remote sensing techniques. Hence, studies to develop techniques that can accurately estimate foliar mineral nutrients are required. Use of the linear models such as PLSR, PCA, or PCR in combination with the non-linear machine learning models has been gaining popularity to retrieve information from the hyperspectral reflectance data [54–56,68]. This can be achieved by using the principal components, latent variables, or variables selected through variable importance [69] as an input for further regression or machine learning modeling. These approaches reduce the collinearity and data dimensionality and increase the computation speed but at the same time retain most of the information of the original dataset [54,68]. The use of linear and non-linear regression analysis has been successfully demonstrated in a few studies but very limited information is available on this aspect in fruit crops, specifically mango.

Mango is the "king of fruits" and its estimated area of cultivation in the world is 5.44 million hectares with production and productivity of 43.3 million tonnes and 7.96 t ha$^{-1}$, respectively [70]. In India, mango is grown over 2.52 million hectares and has a productivity of 6.92 t ha$^{-1}$. The share of India to the world's area of cultivation and production is 46% and 41.6%, respectively. Though the share of area and production of India to the world is huge, the productivity is lesser than the world's average and all other mango-producing countries. In addition to many others, one of the major constraints to the yield is suboptimal and inappropriate nutrient management [71]. Conventional agronomic methods for plant nutrient estimation are being practiced regularly at important growth stages, to manage the fertilizer nutrients [72]. These methods need a huge number of leaf or tissue sample

collection and analysis which is time-consuming and expensive [48,73]. Remote sensing could be a viable tool to estimate the plant's nutritional status and assist in understanding the appropriate amounts of fertilizer inputs in a cost-, labor- and time-effective manner.

Owing to the limited knowledge available on the use of hyperspectral remote sensing to characterize the foliar nutrient status in mango, our study was undertaken with the objectives (1) to compare the efficacy of the spectral indices and chemometric modeling methods and (2) to develop robust quantification models by combining linear and non-linear machine learning models to estimate the foliar macro- and micro-nutrient status of mango.

## 2. Materials and Methods

The objectives of the current study were achieved through four different steps as demonstrated in the scheme of Figure 1. A brief outline about the steps is also presented as follows: Step 1—data collection: field-level leaf sampling and chemical analysis to determine the nutrient content and measurement of the spectral data in the laboratory; Step 2—spectral data pre-processing; Step 3—data analysis: development of vegetation indices and machine learning models, and Step 4—identification of spectral algorithms: identification of the robust spectral algorithms based on the model prediction evaluation parameters.

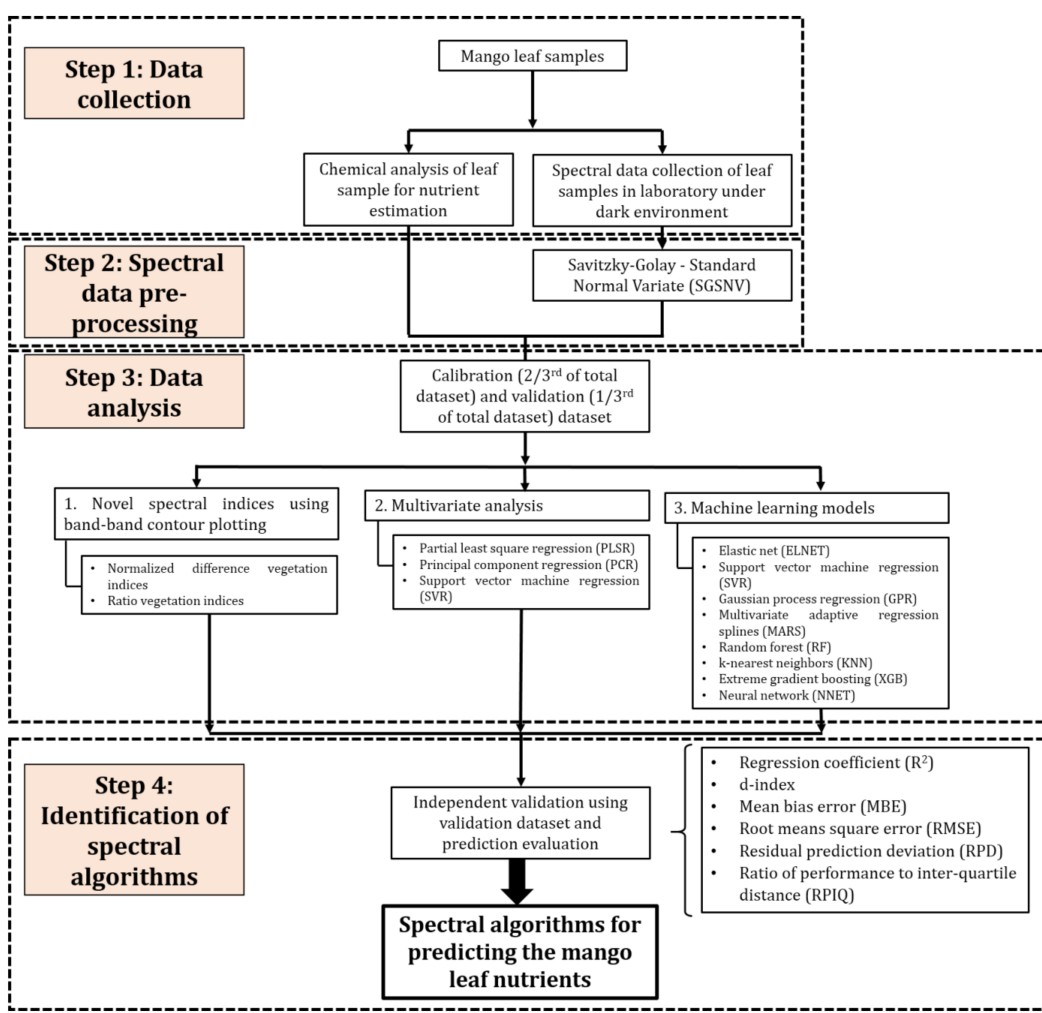

**Figure 1.** The scheme of the methodology followed in the study.

### 2.1. Experimental Setup

Around 400 leaf samples were collected from mango orchards located in North Goa and South Goa districts of Goa State on the west coast of India (Figure 2). Leaf samples were collected from mature plants yielding fruits with an approximate age of 8–10 years. A 4–7-month-old leaf with petiole from the middle of the shoot was collected during the post-fruiting season, i.e., June–July of the years 2018 and 2019. The sampling was undertaken for three weeks. The post-fruiting season was selected in the view that the plant is exhausted of nutrients due to the fruiting in the previous season and gives the actual idea of the nutritional status. The fertilizer application is normally recommended in the second fortnight of June or the end of the monsoon season (second fortnight of September). The post-fruiting stage was ideal for the study to get actual nutritional status and to make the fertilizer prescription for the subsequent season. The samples were collected on dry sunny days.

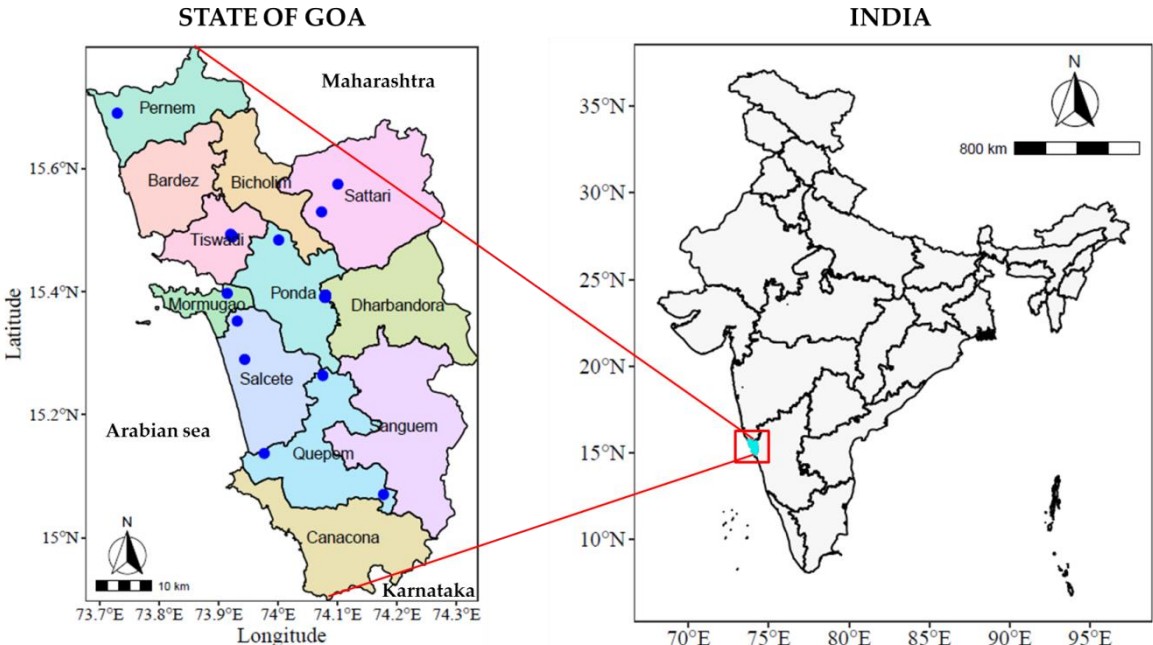

**Figure 2.** Sampling locations of the study. The blue dots on the State of Goa map indicates the locations of orchards of leaf sampling.

### 2.2. Spectral Measurements

A total of 40 orchards were identified for the study, and samples from 10 trees from each orchard were collected, making a total of 400 samples. Immediately after collecting the leaf samples from the field, they were placed in the thermally insulated box to avoid any changes in biochemical properties due to change in temperature and transported to the laboratory. The spectral measurements of the detached leaves were carried out in the laboratory on the same day of leaf sample collection. Mango leaf samples collected were scanned to record the spectral data in the wavelength range of 282–1097 nm using an optical fiber of visible near-infrared spectroradiometer (GER1500, Spectra Vista Corp., Poughkeepsie, NY, USA) as non-contact observations. The Spectroradiometer was calibrated with the Spectralon® panel (Spectra Vista Corp., Poughkeepsie, NY, USA) (100% spectral reflectance) before recording the spectral measurement of the leaf samples. The spectral observations of the adaxial surface of mango leaves were taken within a black box to reduce the impact of stray light. It was ensured that the leaves cover the full field of view of the foreoptics (pistol grip). The spectral observations were taken at nadir position to reduce the impact due to bidirectional reflectance. The calibration was done every after

five samples were recorded. The spectral reflectance data were collected at a bandwidth of 1.5 nm. Further, spectral resampling at a 1 nm interval was done using spline interpolation. The spectral data were further smoothed using Savitzky–Golay filtering across a 15 bands moving window (a window length of 15 and zero polynomial order) to reduce the noise using "prospectr" package in R software version 3.5.2 [74]. The polynomial order was zero. A multiple scatter correction to the data was further done using the standard normal variate technique. Spectral data in the range between 350 nm to 1050 nm was utilized due to the absence of noise. For each leaf sample, an average of five measurements was considered as a representative spectral signature. The average spectral reflectance with standard deviation for the calibration and validation dataset has been presented as Figure 3.

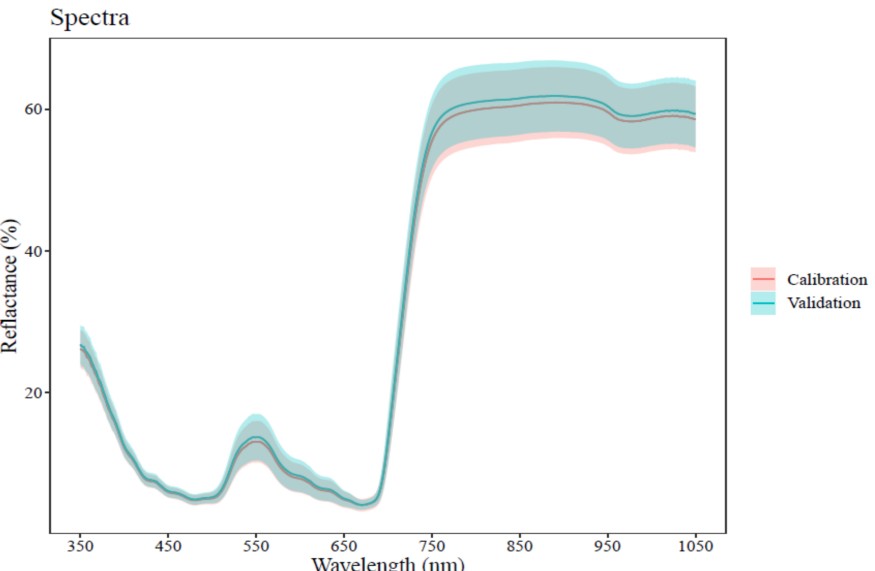

**Figure 3.** The average spectral reflectance with a standard deviation of the calibration and validation dataset (The continuous line is the average reflectance and the shaded area is the standard deviation).

### 2.3. Chemical Analysis

After capturing the leaf spectral data, the samples were oven-dried at 60 °C till constant weight is achieved. The samples were powdered and stored in a zip-lock plastic bag for further chemical analysis. The powdered leaf samples were digested using a mixture of nitric acid and perchloric acid as 9:4 v/v proportion for analysis of the P, K, Ca, Mg, S, Fe, Mn, Zn, Cu, and B [75]. The total N concentration in the mango leaves was estimated using the modified micro Kjeldahls method [76]. Leaf P concentration was determined by measuring the intensity of the yellow color developed by vanado–molybdate reagent with a spectrophotometer [75]. The S concentration of the leaf samples was estimated by measuring turbidity developed by barium chloride using a spectrophotometer [77]. Total leaf K, Ca, Mg, Fe, Mn, Zn, and Cu concentrations were measured with Atomic Absorption Spectrophotometer (nova400P, Analytik Jena, Germany). The B concentration was estimated by measuring a pink color intensity developed in the digest by Azomethine-H indicator with a spectrophotometer. The contents of the nutrients were expressed as percentage and parts per million on a dry weight basis.

Outliers in the 400 nutrient data points were identified and removed resulting in a total of N = 376 sample data for further statistical analysis. The data set was divided into 70% for model calibration (N = 263) and 30% for independent validation (N = 113). The equality of mean, variance, distribution, and CV of calibration and validation datasets were analyzed using t-test, F-test, Kolmogorov–Smirnov test, and Flinger–Kileen test, respectively.



*2.4. Development of Parametric Regression Models*

The best combination of the wavelengths for the development of the VIs was identified using contour plots. The normalized difference spectral index (NDSI) and ratio spectral index (RSI) were calculated as

$$\text{NDSI} = \frac{(R\lambda 1 - R\lambda 2)}{(R\lambda 1 - R\lambda 2)} \tag{1}$$

$$\text{RSI} = \frac{R\lambda 1}{R\lambda 2} \tag{2}$$

Based on all the possible two-pair combination of the wavelengths, the spectral indices were calculated using the software MATLAB. A combination of wavelengths having the highest correlation coefficient with the respective leaf nutrient content was identified for the spectral index.

*2.5. Development of Nonparametric Regression Models*

Multivariate models were built using the hyperspectral reflectance data and corresponding leaf nutrient content. Initially, three different models, i.e., PLSR, PCR, and SVR were tested to retrieve the leaf nutrient contents from the spectral data. The best performing nutrient specific model was identified. The latent variables (LVs) generated from the PLSR model were used as input variables for developing different linear and non-linear models. Machine learning regression algorithms evaluated in the current study were elastic net (ELNET), support vector machine regression (SVR), Gaussian process regression (GPR), multivariate adaptive regression splines (MARS) [78], random forest (RF), k-nearest neighbors (KNN), extreme gradient boosting (XGB) [79], neural network (NNET), and Cubist [80]. The hyper-parameters of each model were calibrated using tenfold cross-validation with five repetitions in "caret" [81] package of R statistical software version 3.5.2 [82]. The hyperparameters which were optimized for each machine learning model were as ELNET-alpha, lambda; SVR-sigma, C; GPR-sigma; MARS-nprune, degree; RF-mtry; splitrule, min.node.size; KNN-Number of neighbors (k); XGB-nrounds, max_depth, eta, gamma, colsample_bytree, min_child_weight, subsample; NNET- size, decay and Cubist -committees, neighbors. Every machine learning model was calibrated using a training dataset using 10-fold cross-validation with five repetitions and thus each model was run 50 times. The performance of a particular model to predict the leaf nutrient content was assessed based on the values of model evaluation parameters such as $R^2$, d-index, mean bias error (MBE), root mean square error (RMSE), residual prediction deviation (RPD), and the ratio of performance to inter-quartile distance (RPIQ). The prediction accuracy of different models was categorized based on RPD as excellent (>2), acceptable ($\geq$1.4–2.0) and non-reliable (<1.40) [83] and RPIQ as very poor (<1.5), poor (1.5–2.0), good (>2.0–2.5) and very good (>2.5) [84]. The values of these parameters are indicated with a superscript letter c and v for calibration and validation, respectively. It is difficult to decide the best-performing model evaluation parameters such as $R^2$, RMSE, RPD, RPIQ, etc. individually. So, a composite summed rank based on these parameters was developed considering the performance of each model parameter wise for the calibration and validation. Ranking of each model evaluation parameter for a particular nutrient in calibration or validation was done using the RANK.AVG function of Microsoft Excel. The ranks of calibration and validation were summed separately and all together and it was referred as a summed rank for a particular model. The model with the least rank predicted the nutrient with the greatest accuracy and the one with the highest rank had the poorest prediction accuracy.

**3. Results**

*3.1. Descriptive Statistics*

The descriptive statistics of the mango leaf nutrients in the full, calibration and validation dataset are presented in Table 1. The coefficient of variation (CV) for the

nutrients analyzed for the calibration and validation dataset varied from 10.30–93.30% and 10.90–88.40%, respectively. For both these datasets, the greatest and least coefficient of variation (CV) was observed for Cu and N, respectively. Similarly, for the full dataset, the greatest (91.90) and least (10.50%) CV was observed for the Cu and N, respectively. All the parameters were positively skewed except for N in full and calibration and N, P, K, and Mg in the validation dataset.

**Table 1.** Summary statistics of the mango leaf nutrient content for full (n = 376), calibration (n = 263) and validation (n = 113) dataset.

| Parameters | N (%) | P (%) | K (%) | Ca (%) | Mg (ppm) | S (%) | Fe (ppm) | Mn (ppm) | Zn (ppm) | Cu (ppm) | B (ppm) |
|---|---|---|---|---|---|---|---|---|---|---|---|
| **Full dataset (n = 376)** | | | | | | | | | | | |
| Minimum | 0.93 | 0.03 | 0.28 | 0.50 | 1477.00 | 0.03 | 57.17 | 8.96 | 11.51 | 0.01 | 12.82 |
| Maximum | 1.47 | 0.18 | 1.72 | 8.13 | 5749.00 | 0.28 | 205.10 | 1959.00 | 26.26 | 1.71 | 102.75 |
| Mean | 1.17 | 0.09 | 0.93 | 3.22 | 3439.63 | 0.15 | 122.88 | 297.67 | 18.14 | 0.51 | 42.18 |
| Standard error | 0.01 | 0.00 | 0.02 | 0.08 | 50.05 | 0.00 | 1.59 | 13.04 | 0.19 | 0.03 | 0.99 |
| Standard deviation | 0.12 | 0.03 | 0.27 | 1.45 | 921.58 | 0.05 | 28.12 | 229.60 | 2.99 | 0.47 | 16.85 |
| Skewness | 0.04 | −0.27 | 0.02 | 0.87 | 0.04 | 0.29 | 0.11 | 2.50 | 0.41 | 0.92 | 0.86 |
| Kurtosis | −1.00 | −0.24 | 0.02 | 0.72 | −0.49 | −0.29 | −0.38 | 11.57 | −0.26 | −0.38 | 0.76 |
| Coefficient of variation (%) | 10.50 | 32.29 | 29.42 | 45.05 | 26.79 | 30.91 | 22.89 | 77.13 | 16.49 | 91.90 | 39.95 |
| **Calibration dataset (n = 263)** | | | | | | | | | | | |
| Minimum | 0.93 | 0.03 | 0.29 | 0.52 | 1559.00 | 0.04 | 57.17 | 8.99 | 12.20 | 0.01 | 14.12 |
| Maximum | 1.47 | 0.18 | 1.72 | 7.93 | 5749.00 | 0.28 | 188.70 | 1959.00 | 26.26 | 1.71 | 102.75 |
| Mean | 1.17 | 0.09 | 0.93 | 3.23 | 3499.93 | 0.15 | 120.68 | 295.93 | 18.40 | 0.52 | 41.33 |
| Standard error | 0.01 | 0.00 | 0.02 | 0.10 | 61.44 | 0.00 | 1.87 | 16.08 | 0.23 | 0.04 | 1.19 |
| Standard deviation | 0.12 | 0.03 | 0.28 | 1.45 | 945.89 | 0.05 | 27.61 | 236.86 | 3.03 | 0.49 | 16.98 |
| Skewness | 0.07 | −0.24 | 0.05 | 0.97 | 0.05 | 0.34 | 0.02 | 2.97 | 0.46 | 0.92 | 1.01 |
| Kurtosis | −0.94 | −0.36 | 0.05 | 0.83 | −0.59 | −0.35 | −0.55 | 14.66 | −0.31 | −0.43 | 1.14 |
| Coefficient of variation (%) | 10.35 | 32.16 | 29.85 | 44.87 | 27.03 | 31.03 | 22.88 | 80.04 | 16.48 | 93.30 | 41.07 |
| **Validation dataset (n = 113)** | | | | | | | | | | | |
| Minimum | 0.93 | 0.03 | 0.28 | 0.50 | 1477.00 | 0.03 | 60.60 | 8.96 | 11.51 | 0.01 | 12.82 |
| Maximum | 1.39 | 0.17 | 1.59 | 8.13 | 5470.00 | 0.27 | 205.10 | 968.60 | 25.15 | 1.55 | 96.25 |
| Mean | 1.17 | 0.09 | 0.94 | 3.19 | 3299.52 | 0.15 | 127.99 | 301.71 | 17.55 | 0.49 | 44.12 |
| Standard error | 0.01 | 0.00 | 0.03 | 0.15 | 84.21 | 0.00 | 2.97 | 22.08 | 0.32 | 0.05 | 1.76 |
| Standard deviation | 0.13 | 0.03 | 0.27 | 1.46 | 850.46 | 0.05 | 28.78 | 212.89 | 2.83 | 0.43 | 16.49 |
| Skewness | −0.01 | −0.33 | −0.06 | 0.65 | −0.12 | 0.15 | 0.25 | 0.97 | 0.25 | 0.85 | 0.53 |
| Kurtosis | −1.11 | 0.11 | 0.04 | 0.55 | −0.36 | −0.12 | −0.21 | 0.52 | −0.40 | −0.36 | 0.16 |
| Coefficient of variation (%) | 10.90 | 32.67 | 28.57 | 45.70 | 25.78 | 30.77 | 22.49 | 70.56 | 16.10 | 88.46 | 37.37 |
| *p*-value | | | | | | | | | | | |
| *t*-test | 0.81 | 0.49 | 0.81 | 0.81 | 0.07 | 0.71 | 0.06 | 0.84 | 0.06 | 0.63 | 0.20 |
| F-test | 0.54 | 0.92 | 0.73 | 0.93 | 0.22 | 0.82 | 0.62 | 0.24 | 0.49 | 0.27 | 0.77 |
| Kolmogorov-Smirnov test | 0.89 | 0.03 | 0.89 | 0.85 | 0.15 | 0.90 | 0.42 | 0.76 | 0.14 | 0.76 | 0.15 |
| Flinger-Kileen test | 0.22 | 0.47 | 0.40 | 0.16 | 0.34 | 0.48 | 0.30 | 0.09 | 0.49 | 0.32 | 0.48 |

N, nitrogen; P, phosphorus; K, potassium; Ca, calcium; Mg, magnesium; S, sulfur; Fe, iron; Mn, manganese; Zn, zinc; Cu, copper; B, boron.

The results revealed that the difference between calibration and validation dataset for mean, variance, and CV was insignificant. Kolmogorov–Smirnov test showed an equal distribution of leaf nutrient content across the calibration and validation dataset ($p > 0.05$) except for P ($p = 0.03$). These results confirm that the calibration and validation dataset are statistically similar and the random selection employed is appropriate. The calibration and validation dataset represented the variability present in the full dataset. The Jarque–Bera test of normality indicated that all the parameters were normally distributed except N, Ca, Mn, Zn, Cu, and B (Table 1). The values of these nutrients were Box–Cox transformed to make them normally distributed before the data were employed for further statistical analysis (Table 2).

**Table 2.** Jarque–Bera test of normality for raw and transformed data.

| Variables | Raw Data | | Box-Cox Lambda | Transformed Data | |
|---|---|---|---|---|---|
| | Jarque-Bera | *p*-Value | | Jarque-Bera | *p*-Value |
| N | 11.36 | 0.003 | 0.59 | 11.45 | 0.01 |
| P | 5.03 | 0.081 | | | |
| K | 0.01 | 0.993 | | | |
| Ca | 44.66 | $2.00 \times 10^{-10}$ | 0.38 | 0.54 | 0.75 |
| Mg | 3.65 | 0.161 | | | |
| S | 5.30 | 0.071 | | | |
| Fe | 2.57 | 0.277 | | | |
| Mn | 1988.00 | 0.000 | 0.27 | 4.31 | 0.09 |
| Zn | 8.09 | 0.018 | −0.09 | 2.18 | 0.28 |
| Cu | 36.57 | $1.15 \times 10^{-8}$ | −1.37 | 18.34 | 0.00 |
| B | 41.71 | $8.75 \times 10^{-10}$ | 0.14 | 1.49 | 0.44 |

N, nitrogen; P, phosphorus; K, potassium; Ca, calcium; Mg, magnesium; S, sulfur; Fe, iron; Mn, manganese; Zn, zinc; Cu, copper; B, boron.

### 3.2. Indices Development and Prediction Performance

The best combinations of the wavelengths for the development of the VIs were identified through contour plots were generated and are presented in Figures 4 and 5. The NDSI and RSI identified for each nutrient have been listed in Table 3, with the results of the prediction performance. In the case of NDSI, for calibration, the prediction accuracy as indicated by $R^2_c$ ranged from 0.002 (N) to 0.466 (Mg) with $RMSE_c$ of 0.09 (S) to 2672.08 (Mg), respectively, while for validation, the $R^2_v$ varied from 0.05 (N) to 0.41 (K) with $RMSE_v$ of 0.11 (P) to 2739.48 (Mg). In general, for calibration and validation, the RPD and RPIQ values were $\leq 0.94$ and $\leq 1.25$. It indicated that the predictions were very poor for all the nutrients using the NDSIs. In the case of RSIs, the $R^2$ and RMSE varied from 0.04 (N) to 0.50 (Mg) and 0.11(P) to 871.58 (Mg), respectively, during calibration and from 0.06 (Cu)–0.38 (Ca) and 0.08 (S)−1081.80 (Mg), respectively, for validation. For the calibration and validation prediction using the RSIs, the RPD and RPIQ were $\leq 1.03$ and $\leq 1.35$. Similar to the NDSIs, the predictions using the RSIs were also very poor. In the current study, none of the spectral indices developed could yield successful predictions for any of the nutrients.

### 3.3. Performance of Nonparametric Regression Analysis

Multivariate analysis techniques such as PLSR, PCR, and SVR were employed to predict the mango leaf nutrient contents using the calibration and validation dataset, respectively (Table 4). Model evaluation parameters such as $R^2$, d-index, MBE, RMSE, RPD, and RPIQ were used to evaluate the prediction accuracy of the model. To avoid the complexity of deciding the performance of the model using the model evaluation parameters individually, a composite rank based on these parameters was developed considering the performance during calibration and validation. Overall sum ranking showed that the PLSR model was the best to predict most of the nutrients except N, S (best obtained by SVR), and Mn (best obtained by PCR) in which predictions were unreliable. The accuracy of the PLSR to predict P, K, Ca, Mg, Fe, Mn, Zn, and B with respect to $R^2_c$, $RPD_c$, and $RPIQ_c$ for calibration varied from 0.34–0.59, 1.23–1.56, and 1.47–2.13. During validation, these indices ranged from 0.26–0.53, 1.11–1.42, and 1.34–1.79, respectively. The greatest prediction accuracy was achieved for the leaf Ca ($R^2_v$ = 0.53, $RPD_v$ = 1.42 and $RPIQ_v$ = 1.79 for the independent validation). Based on the RPIQ values for both calibration and validation, predictions for all the nutrients were categorized as poor except for Mg during calibration (RPIQ = 2.13, very good). However, as per the criteria of RPD, the P, K, Ca, and Mg, predictions for calibration and Ca and Mg for validation were acceptable. Among all the nutrients, the performance of these three multivariate models to predict N and Cu was the poorest with $R^2 \leq 0.19$, $RPD \leq 1.10$, and $RPIQ \leq 1.90$, indicating very poor prediction for both calibration and validation.

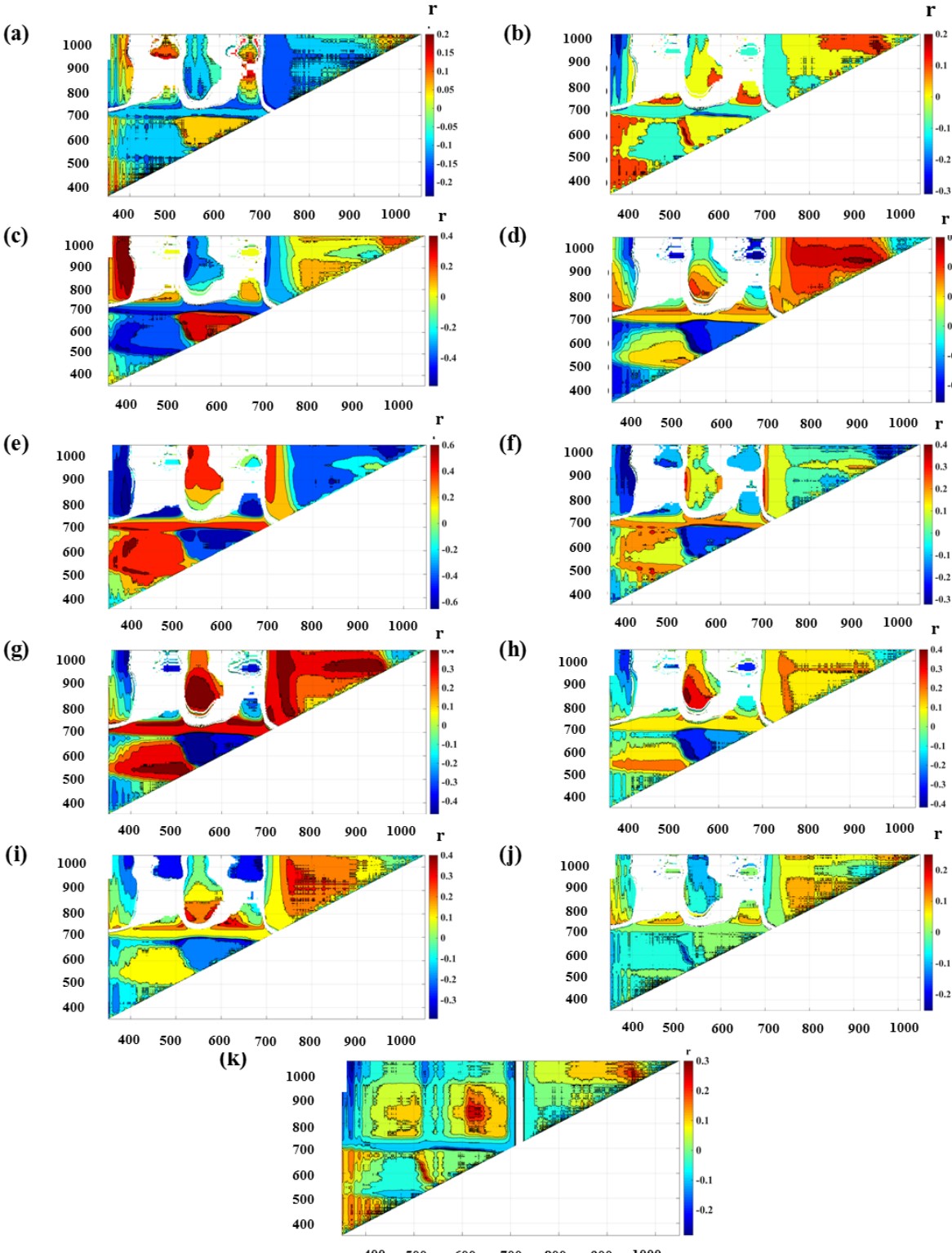

**Figure 4.** Contour plots for the linear relationship of normalized difference spectral indices with (**a**) N, (**b**) P, (**c**) K, (**d**) Ca, (**e**) Mg, (**f**) S, (**g**) Fe, (**h**) Mn, (**i**) Zn, (**j**) Cu, and (**k**) B. (Both x and y-axis represent wavelength in nm, and r is the correlation coefficient).

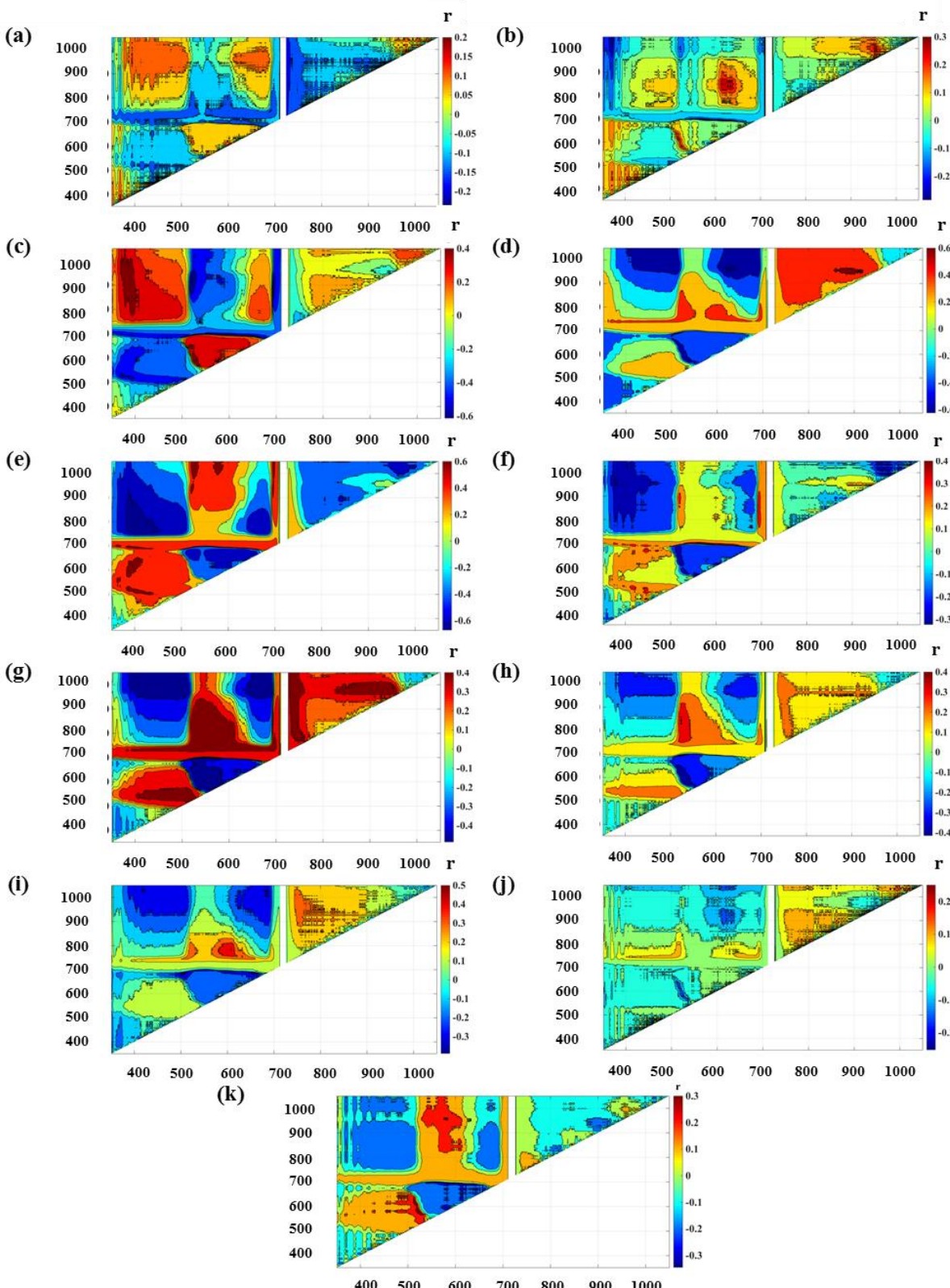

**Figure 5.** Contour plots for the linear relationship of ratio spectral indices with (**a**) N, (**b**) P, (**c**) K, (**d**) Ca, (**e**) Mg, (**f**) S, (**g**) Fe, (**h**) Mn, (**i**) Zn, (**j**) Cu, and (**k**) B. (Both x and y-axis represent wavelength in nm, and r is the correlation coefficient).

**Table 3.** Performance of the best-identified normalized difference and ratio vegetation indices to predict the leaf nutrient content.

| Nutrients | Vegetation Index | Index Formula | Calibration | | | | | | Validation | | | | | |
|---|---|---|---|---|---|---|---|---|---|---|---|---|---|---|
| | | | $R^2_c$ | $Dindex_c$ | $MBE_c$ | $RMSE_c$ | $RPD_c$ | $RPIQ_c$ | $R^2_v$ | $Dindex_v$ | $MBE_v$ | $RMSE_v$ | $RPD_v$ | $RPIQ_v$ |
| | | | Normalized difference spectral indices | | | | | | | | | | | |
| N | ND_685_941 | $\frac{R941-R685}{R941+R685}$ | 0.002 | 0.06 | 0.00 | 2.60 | 0.05 | 0.08 | 0.05 | 0.05 | 0.99 | 3.72 | 0.03 | 0.05 |
| P | ND_957_988 | $\frac{R988-R957}{R988+R957}$ | 0.073 | 0.36 | 0.00 | 0.11 | 0.28 | 0.37 | 0.13 | 0.35 | -0.01 | 0.11 | 0.24 | 0.27 |
| K | ND_522_914 | $\frac{R914-R522}{R914-R522}$ | 0.356 | 0.72 | 0.00 | 0.37 | 0.75 | 0.93 | 0.41 | 0.77 | 0.01 | 0.30 | 0.94 | 1.25 |
| Ca | ND_883_956 | $\frac{R956-R883}{R956+R883}$ | 0.412 | 0.76 | 0.03 | 1.82 | 0.83 | 0.99 | 0.38 | 0.74 | 0.33 | 1.62 | 0.80 | 0.97 |
| Mg | ND_388_806 | $\frac{R806-R388}{R806+R388}$ | 0.466 | 0.45 | 2214.95 | 2672.08 | 0.34 | 0.44 | 0.40 | 0.47 | 2167.71 | 2739.48 | 0.36 | 0.49 |
| S | ND_578_697 | $\frac{R697-R578}{R697+R578}$ | 0.209 | 0.57 | 0.00 | 0.09 | 0.51 | 0.68 | 0.30 | 0.65 | 0.00 | 0.08 | 0.59 | 0.89 |
| Fe | ND_531_863 | $\frac{R863-R531}{R863+R531}$ | 0.218 | 0.56 | 10.09 | 58.68 | 0.48 | 0.69 | 0.28 | 0.57 | 5.03 | 59.01 | 0.47 | 0.57 |
| Mn | ND_524_848 | $\frac{R848-R524}{R848+R524}$ | 0.218 | 0.58 | −12.36 | 444.20 | 0.55 | 0.60 | 0.08 | 0.45 | −47.78 | 434.64 | 0.44 | 0.53 |
| Zn | ND_611_760 | $\frac{R760-R611}{R760+R611}$ | 0.162 | 0.50 | 0.11 | 6.53 | 0.44 | 0.55 | 0.31 | 0.63 | 0.23 | 5.96 | 0.55 | 0.88 |
| Cu | ND_842_853 | $\frac{R853-R842}{R853+R842}$ | 0.065 | 0.35 | 0.02 | 1.72 | 0.27 | 0.43 | 0.06 | 0.36 | 0.34 | 1.77 | 0.27 | 0.37 |
| B | ND_512_615 | $\frac{R615-R512}{R615+R512}$ | 0.078 | 0.26 | 17.56 | 87.05 | 0.20 | 0.25 | 0.15 | 0.26 | 6.01 | 99.04 | 0.16 | 0.24 |
| | | | Ratio spectral indices | | | | | | | | | | | |
| N | R_927_932 | $\frac{R932}{R927}$ | 0.04 | 0.31 | 0.01 | 0.60 | 0.21 | 0.37 | 0.10 | 0.29 | 0.18 | 0.68 | 0.18 | 0.25 |
| P | R_615_849 | $\frac{R849}{R615}$ | 0.07 | 0.36 | 0.00 | 0.11 | 0.28 | 0.37 | 0.17 | 0.39 | 0.01 | 0.10 | 0.27 | 0.30 |
| K | R_522_925 | $\frac{R925}{R522}$ | 0.38 | 0.74 | 0.00 | 0.35 | 0.78 | 0.98 | 0.37 | 0.75 | 0.02 | 0.30 | 0.93 | 1.24 |
| Ca | R_883_956 | $\frac{R956}{R883}$ | 0.41 | 0.76 | 0.00 | 1.80 | 0.84 | 1.00 | 0.38 | 0.74 | 0.29 | 1.60 | 0.81 | 0.98 |
| Mg | R_525_1026 | $\frac{R1026}{R525}$ | 0.50 | 0.81 | −102.48 | 871.58 | 1.03 | 1.35 | 0.34 | 0.74 | −127.20 | 1081.80 | 0.91 | 1.24 |
| S | R_578_697 | $\frac{R697}{R578}$ | 0.21 | 0.57 | 0.00 | 0.09 | 0.51 | 0.68 | 0.30 | 0.65 | 0.00 | 0.08 | 0.59 | 0.89 |
| Fe | R_531_842 | $\frac{R842}{R531}$ | 0.23 | 0.60 | −1.22 | 50.64 | 0.56 | 0.80 | 0.29 | 0.60 | -6.63 | 53.09 | 0.52 | 0.64 |
| Mn | R_522_848 | $\frac{R848}{R522}$ | 0.22 | 0.55 | 35.71 | 506.37 | 0.48 | 0.52 | 0.07 | 0.41 | 16.10 | 481.53 | 0.40 | 0.48 |
| Zn | R_608_780 | $\frac{R780}{R608}$ | 0.23 | 0.60 | −0.62 | 5.01 | 0.57 | 0.72 | 0.37 | 0.73 | 0.06 | 4.05 | 0.81 | 1.29 |
| Cu | R_842_853 | $\frac{R853}{R842}$ | 0.07 | 0.34 | 0.00 | 1.79 | 0.26 | 0.41 | 0.06 | 0.35 | 0.34 | 1.83 | 0.26 | 0.36 |
| B | R_515_615 | $\frac{R615}{R515}$ | 0.31 | 0.63 | 0.23 | 5.96 | 0.55 | 0.88 | 0.19 | 0.37 | −1.60 | 65.18 | 0.24 | 0.37 |

N, nitrogen; P, phosphorus; K, potassium; Ca, calcium; Mg, magnesium; S, sulfur; Fe, iron; Mn, manganese; Zn, zinc; Cu, copper; B, boron; $R^2$, regression coefficient; MBE, mean bias error; RMSE, root means square error; RPD, ratio of performance to deviation; RPIQ, ratio of performance to inter-quartile distance.

### 3.4. Performance of the PLSR-Combined Machine Learning Models

The results pertaining to the prediction performance of the PLSR-combined machine learning models are presented in Table A1 and the best performing models in Figure 6. The optimum number of latent variables (LVs) were generated and selected using PLSR and 10-fold cross-validation and used as predictor variables for machine learning model development. The LVs are linear combinations of all the input variables but orthogonal to each other which helps to reduce collinearity. The number of the LVs for N, P, K, Ca, Mg, S, Fe, Mn, Zn, Cu, and B selected were 1, 6, 5, 5, 7, 10, 5, 5, 5, 1, and 6, respectively. Overall, the prediction performance improved significantly with the PLSR-combined machine learning models over the single PLSR model. For all the nutrients, the performance of the best performing PLSR-combined machine learning models with respect to $R^2$, RPD, and RPIQ for calibration ranged from 0.95 to 0.99, 4.42 to 11.06, and 6.55–13.80, and for validation, these were 0.88–0.99, 2.73 to 5.76, and 3.31 to 7.65, respectively. Based on the RPD and RPIQ values, it was evident that all the machine learning models combined with PLSR were effective to predict all the macro- and micro-nutrients with very good to excellent prediction accuracy. Based on the independent validation performance and the summed ranks, the best performing model for different nutrients were Cubist for N ($R^2_v$ = 0.94, $RPD_v$ = 4.27, and $RPIQ_v$ = 6.03), P ($R^2_v$ = 0.91, $RPD_v$ = 3.3, and $RPIQ_v$ = 3.71), K ($R^2_v$ = 0.97, $RPD_v$ = 5.76, and $RPIQ_v$ = 7.65,), and Zn ($R^2_v$ = 0.95, $RPD_v$ = 4.71, and $RPIQ_v$ = 7.49), SVR for Ca ($R^2_v$ = 0.88, $RPD_v$ = 2.73, and $RPIQ_v$ = 3.31), Fe ($R^2_v$ = 0.91, $RPD_v$ = 3.26, and $RPIQ_v$ = 3.96), Cu ($R^2_v$ = 0.90, $RPD_v$ = 3.01, and $RPIQ_v$ = 3.82), and B ($R^2_v$ = 0.92, $RPD_v$ = 3.44, and $RPIQ_v$ = 5.33), and ELNET for Mg ($R^2_v$ = 0.95, $RPD_v$ = 4.47, and $RPIQ_v$ = 6.11) and S ($R^2_v$ = 0.95, $RPD_v$ = 4.48, and $RPIQ_v$ = 6.8). Although the prediction accuracies for all the models were very good to excellent, the most robust PLSR-combined models were Cubist, SVR, and ELNET. Table 5 gives an overview of the independent validation performance and identifies the best machine learning models combined with PLSR to predict mango leaf nutrients based on the RPD and RPIQ. Among the nine machine learning models tested, the performance of the MARS, RF, and KNN was the poorest and yielded non-reliable predictions for most of the nutrients except MARS for K and Zn and KNN for Mg and B.

**Table 4.** Performance of the multivariate analysis models for calibration and validation to predict the mango leaf nutrient content.

| Model | Calibration | | | | | | Validation | | | | | | Summed Rank |
|---|---|---|---|---|---|---|---|---|---|---|---|---|---|
| | $R^2_c$ | $dindex_c$ | $MBE_c$ | $RMSE_c$ | $RPD_c$ | $RPIQ_c$ | $R^2_v$ | $dindex_v$ | $MBE_v$ | $RMSE_v$ | $RPD_v$ | $RPIQ_v$ | |
| **N** | | | | | | | | | | | | | |
| PLSR | 0.031 | 0.231 | $4.76 \times 10^{-8}$ | 0.12 | 1.02 | 1.77 | 0.001 | 0.184 | 0.00297 | 0.13 | 0.98 | 1.61 | 24 |
| PCR | 0.028 | 0.222 | $4.76 \times 10^{-8}$ | 0.12 | 1.02 | 1.76 | 0.001 | 0.176 | 0.0026 | 0.13 | 0.98 | 1.62 | 25 |
| SVR | 0.171 | 0.481 | 0.003614 | 0.11 | 1.10 | 1.90 | 0.006 | 0.327 | 0.01311 | 0.13 | 0.98 | 1.61 | 22 |
| **P** | | | | | | | | | | | | | |
| PLSR | 0.337 | 0.695 | $-1.45 \times 10^{-8}$ | 0.02 | 1.23 | 1.47 | 0.404 | 0.747 | 0.0003 | 0.02 | 1.30 | 1.34 | 14 |
| PCR | 0.250 | 0.616 | $1.13 \times 10^{-9}$ | 0.03 | 1.16 | 1.38 | 0.386 | 0.697 | −0.001 | 0.02 | 1.27 | 1.31 | 25 |
| SVR | 0.265 | 0.533 | 0.00213 | 0.03 | 1.14 | 1.36 | 0.458 | 0.609 | 0.0026 | 0.02 | 1.24 | 1.28 | 33 |
| **K** | | | | | | | | | | | | | |
| PLSR | 0.543 | 0.836 | $6.86 \times 10^{-8}$ | 0.19 | 1.48 | 1.91 | 0.431 | 0.794 | −0.0051 | 0.20 | 1.32 | 1.77 | 14 |
| PCR | 0.222 | 0.589 | $1.24 \times 10^{-8}$ | 0.24 | 1.14 | 1.46 | 0.289 | 0.616 | 0.00131 | 0.23 | 1.19 | 1.59 | 32 |
| SVR | 0.480 | 0.707 | −0.00827 | 0.21 | 1.33 | 1.71 | 0.349 | 0.654 | −0.013 | 0.22 | 1.23 | 1.66 | 26 |
| **Ca** | | | | | | | | | | | | | |
| PLSR | 0.518 | 0.824 | $-4.86 \times 10^{-7}$ | 1.00 | 1.44 | 1.67 | 0.485 | 0.804 | −0.0768 | 1.04 | 1.40 | 1.79 | 19 |
| PCR | 0.483 | 0.806 | $-1.61 \times 10^{-7}$ | 1.04 | 1.39 | 1.62 | 0.487 | 0.795 | −0.0635 | 1.04 | 1.40 | 1.79 | 22 |
| SVR | 0.538 | 0.801 | −0.13348 | 1.01 | 1.44 | 1.67 | 0.434 | 0.717 | −0.2413 | 1.13 | 1.29 | 1.65 | 31 |
| **Mg** | | | | | | | | | | | | | |
| PLSR | 0.588 | 0.855 | −0.00073 | 605.77 | 1.56 | 2.13 | 0.527 | 0.838 | 72.5333 | 597.27 | 1.42 | 1.76 | 17 |
| PCR | 0.412 | 0.753 | $-9.70 \times 10^{-5}$ | 724.09 | 1.31 | 1.78 | 0.423 | 0.759 | 180.961 | 668.44 | 1.27 | 1.57 | 34 |
| SVR | 0.504 | 0.794 | −2.10901 | 668.49 | 1.41 | 1.93 | 0.550 | 0.824 | 152.705 | 588.85 | 1.44 | 1.79 | 21 |
| **S** | | | | | | | | | | | | | |
| PLSR | 0.256 | 0.626 | $-5.71 \times 10^{-9}$ | 0.04 | 1.16 | 1.48 | 0.177 | 0.561 | −0.0002 | 0.04 | 1.11 | 1.44 | 30 |
| PCR | 0.312 | 0.672 | $-6.05 \times 10^{-9}$ | 0.04 | 1.21 | 1.54 | 0.195 | 0.593 | −0.0003 | 0.04 | 1.12 | 1.46 | 22 |
| SVR | 0.371 | 0.604 | −0.003 | 0.04 | 1.21 | 1.54 | 0.215 | 0.506 | −0.0025 | 0.04 | 1.12 | 1.46 | 20 |
| **Fe** | | | | | | | | | | | | | |
| PLSR | 0.483 | 0.803 | $4.57 \times 10^{-6}$ | 19.81 | 1.39 | 1.94 | 0.324 | 0.691 | −4.6565 | 24.00 | 1.20 | 1.69 | 13 |
| PCR | 0.278 | 0.649 | $2.15 \times 10^{-6}$ | 23.42 | 1.18 | 1.64 | 0.220 | 0.567 | −8.8722 | 26.80 | 1.07 | 1.52 | 33 |
| SVR | 0.405 | 0.737 | −1.18679 | 21.33 | 1.29 | 1.80 | 0.323 | 0.601 | −8.9331 | 25.59 | 1.12 | 1.59 | 26 |
| **Mn** | | | | | | | | | | | | | |
| PLSR | 0.306 | 0.680 | $-2.44 \times 10^{-5}$ | 196.80 | 1.20 | 1.22 | 0.166 | 0.596 | −18.421 | 198.96 | 1.07 | 1.47 | 27 |
| PCR | 0.307 | 0.678 | $1.06 \times 10^{-5}$ | 196.76 | 1.20 | 1.22 | 0.181 | 0.612 | −23.217 | 197.48 | 1.08 | 1.48 | 19 |
| SVR | 0.340 | 0.526 | −29.8089 | 204.04 | 1.16 | 1.18 | 0.211 | 0.519 | −48.979 | 195.44 | 1.09 | 1.49 | 26 |
| **Zn** | | | | | | | | | | | | | |
| PLSR | 0.417 | 0.758 | $-1.68 \times 10^{-7}$ | 2.31 | 1.31 | 1.85 | 0.284 | 0.677 | 0.45319 | 2.43 | 1.16 | 1.56 | 14 |
| PCR | 0.365 | 0.721 | $2.23 \times 10^{-7}$ | 2.41 | 1.26 | 1.78 | 0.234 | 0.636 | 0.40979 | 2.50 | 1.13 | 1.52 | 25 |
| SVR | 0.387 | 0.666 | −0.24371 | 2.43 | 1.25 | 1.76 | 0.114 | 0.462 | 0.28452 | 2.66 | 1.06 | 1.43 | 33 |
| **Cu** | | | | | | | | | | | | | |
| PLSR | 0.011 | 0.142 | $1.81 \times 10^{-8}$ | 0.48 | 1.01 | 1.62 | 0.020 | 0.160 | 0.02837 | 0.43 | 1.01 | 1.51 | 20 |
| PCR | 0.010 | 0.004 | $1.47 \times 10^{-8}$ | 0.48 | 1.00 | 1.61 | 0.001 | 0.106 | 0.03061 | 0.43 | 1.00 | 1.50 | 30 |
| SVR | 0.199 | 0.442 | −0.15908 | 0.48 | 1.02 | 1.64 | 0.098 | 0.402 | −0.135 | 0.43 | 1.00 | 1.50 | 22 |
| **B** | | | | | | | | | | | | | |
| PLSR | 0.375 | 0.725 | $-2.91 \times 10^{-6}$ | 13.39 | 1.27 | 1.61 | 0.258 | 0.681 | −3.1827 | 14.88 | 1.11 | 1.54 | 18 |
| PCR | 0.319 | 0.686 | $-3.79 \times 10^{-6}$ | 13.97 | 1.22 | 1.54 | 0.294 | 0.694 | −3.1775 | 14.28 | 1.15 | 1.60 | 19 |
| SVR | 0.359 | 0.511 | −2.27752 | 14.67 | 1.16 | 1.47 | 0.210 | 0.492 | −5.242 | 15.67 | 1.05 | 1.46 | 35 |

N, nitrogen; P, phosphorus; K, potassium; Ca, calcium; Mg, magnesium; S, sulfur; Fe, iron; Mn, manganese; Zn, zinc; Cu, copper; B, boron; R2, regression coefficient; MBE, mean bias error; RMSE, root means square error; RPD, ratio of performance to deviation; RPIQ, ratio of performance to inter-quartile distance; PLSR, partial least square regression; PCR, principal component regression; SVR, support vector regression.

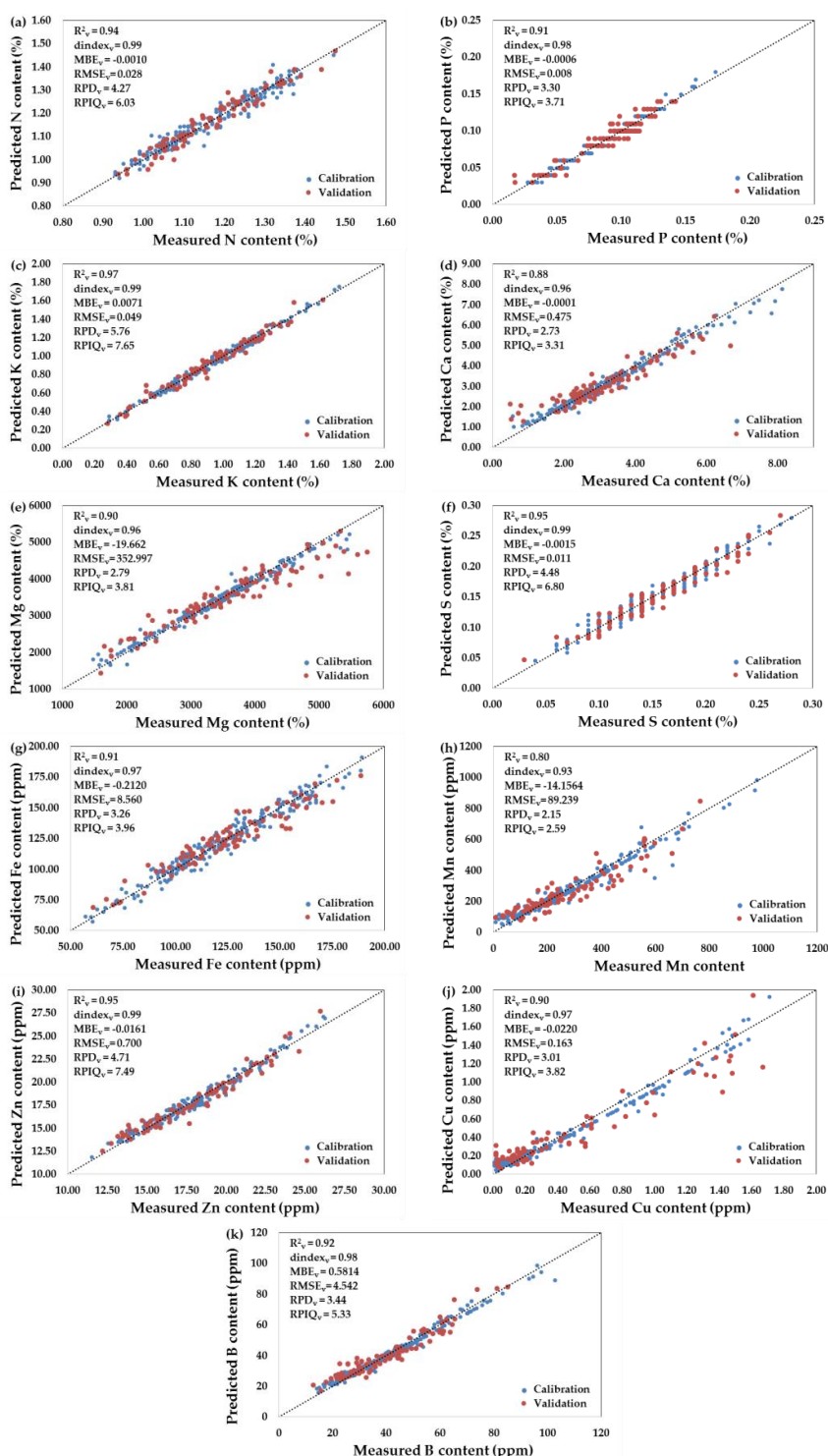

**Figure 6.** Performance of the best performing PLSR-combined models for predicting nutrients as (**a**) N using PLSR-Cubist, (**b**) P using PLSR-Cubist, (**c**) K using PLSR-Cubist, (**d**) Ca using PLSR-SVR, (**e**) Mg using PLSR-SVR, (**f**) S using PLSR elastic net (ELNET), (**g**) Fe using PLSR-SVR, (**h**) Mn using PLSR-SVR, (**i**) Zn using PLSR-Cubist, (**j**) Cu using PLSR-SVR, and (**k**) B using PLSR-SVR.

**Table 5.** Summarized independent validation performance of the machine learning models in combination with PLSR for predicting mango leaf nutrients based on RPD and RPIQ.

| Model | Based on RPD (>2): Excellent Prediction Accuracy | | | | | | | | | | | Based on RPIQ (>2.5): Very Good Prediction Accuracy | | | | | | | | | | |
|---|---|---|---|---|---|---|---|---|---|---|---|---|---|---|---|---|---|---|---|---|---|---|
| | N | P | K | Ca | Mg | S | Fe | Mn | Zn | Cu | B | N | P | K | Ca | Mg | S | Fe | Mn | Zn | Cu | B |
| ELNET | × | × | × | × | × | × | × | × | × | × | × | × | × | × | × | × | × | × | × | × | × | × |
| SVR | × | × | × | × | × | × | × | × | × | × | × | × | × | × | × | × | × | × | × | × |  | × |
| GPR | × | × | × | × | × | × | × |  | × | × | × | × | × | × | × | × | × | × |  | × |  | × |
| MARS |  |  | × |  |  |  |  |  |  |  |  |  |  | × |  |  |  |  |  | × |  |  |
| RF |  |  |  |  |  |  |  |  |  |  |  |  |  |  |  |  |  |  |  |  |  |  |
| KNN |  |  |  |  |  |  |  |  |  |  |  |  |  |  |  | × |  |  |  |  |  | × |
| XGB | × | × | × | × | × | × | × |  | × |  |  | × | × | × | × | × | × | × |  | × |  | × |
| NNET | × | × | × | × |  | × |  |  | × | × | × | × | × | × | × | × |  | × |  | × | × | × |
| Cubist | × | × | × | × | × | × | × | × | × | × | × | × | × | × | × | × | × | × | × | × | × | × |

×, indicates excellent and very good predictions using ratio to performance (RPD) and ratio of performance to interquartile distance (RPIQ), respectively. N, nitrogen; P, phosphorus; K, potassium; Ca, calcium; Mg, magnesium; S, sulfur; Fe, iron; Mn, manganese; Zn, zinc; Cu, copper; B, boron; ELNET, elastic net; SVR, support vector regression; GPR, Gaussian process regression; MARS, multivariate adaptive regression splines; RF, random forest; KNN, k-nearest neighbors; XGB, extreme gradient boosting; NNET, neural network.

## 4. Discussion

### 4.1. Variations in Leaf Nutrient Concentrations and Spectral Data

Before employing the nutrient and spectral data for the statistical analysis, it is very important to appropriately split the data into calibration and validation datasets. The insignificant results of the t-test, F-test, Kolmogorov–Smirnov test, and Flinger–Kileen test indicated that the random division of the data into calibration and validation datasets was accurate, rendering it suitable for further statistical analysis. The variations in the spectral data were more prominent in the NIR region than in the visible. Similar findings were noted by [85] and [20] while predicting the leaf ion content using remote sensing in cotton and rice, respectively. The spectral pattern and variations recorded made the spectral data suitable for further analysis. Prominent leaf spectral variations in visible-NIR regions were reported by [21] for predicting macro- and micro-nutrient content in orange. [49] showed the hyperspectral features in the spectral region of 470–800 nm are useful for detecting concentrations of leaf nutritional elements. In [86], variations in the spectral signature of oil palm for different nutrient such as N, P, K, Mg, Ca, and B were observed. Higher reflectance in the infrared region (650–900 nm) was also observed by [87] in groundnut plants while predicting the N, P, and K content and yield. Wide variation in the nutrient status of the plant is an important pre-requisite to developing prediction models from remote sensing data. In the current study, a wide variation in the nutrient data was observed for all except for N and Zn. Such observations are supported by the results obtained by [49] with the highest variations for Ca and least for Mg in tallgrass prairie vegetation. The degree of variation may also affect the prediction of nutrients using spectral data and different statistical analysis methods.

### 4.2. Vegetation Indices

In the current study, none of the spectral indices developed could predict any of the leaf nutrients successfully. The inability of model development by spectral indices could be the outcome of an unsuccessful match of selected indices and wavelengths as individual wavelengths and/or regions might not have strongly correlated with nutrient concentrations. Another probable explanation could be the inability to better deal with confounding factors such as reflectance saturation, leaf area, roughness, and moisture in the leaf, which reduces the performance of raw spectral bands [88]. Earlier studies used different vegetation indices for predicting foliar nutrient in different crops and most of these were used to detect foliar or canopy N, P, and K content as they are powerful indicators of plant nutrition status [89,90]. Normalized difference spectral indices were effectively used by [91] and [92] to estimate leaf N, P, or K content in different plant species.

In [11], a poor prediction of already published 43 empirical spectral indices for the N, P, and K content of the shrub and grass vegetation in China was recorded. Furthermore, to overcome this, the linear regression analysis to optimize the band-band combination was performed and effectively retrieved the leaf N, P, and K content ($R^2 > 0.5$, $p < 0.05$). This confirms that hyperspectral data could be potentially used for fine-scale monitoring of degraded vegetation.

The use of few wavelengths to develop a spectral index and for the prediction of nutrients offers a simple way to model any parameter, but at the same time, does not consider the information hidden in the other parts of the spectrum or wavelengths. A poor prediction accuracy was found by [93] exploiting the Inverted red-edge chlorophyll index ($R^2 = 0.66$), relative normalized difference index ($R^2 = 0.48$), red-edge chlorophyll index ($R^2 = 0.28$), and normalized difference infrared index ranged $R^2 = 0.28-0.67$ for the coffee canopy N using satellite data. Thus, our results on the poor performance of spectral or vegetation indices are consistent with those reported by [11,93], among others. Prediction accuracy of $R^2 = 0.16-0.48$ was obtained by [94] to predict the N:P ratio of the grass vegetation using previously published vegetation indices computed from the satellite data however the performance was improved to $R^2 = 0.59-0.72$ with optimized vegetation indices.

### 4.3. Chemometrics and Machine Learning Regression Modeling

Among the three multivariate models tested, the PLSR was the best to predict most of the nutrients except SVR for N, S, and PCR for Mn. However, the prediction performance was poor with $R^2 \leq 0.53$ and low values of RPD and RPIQ (Table 4). [95] and [96] predicted the leaf nutritional elements with PLSR modeling of the spectral data. Our results are also consistent with that of [49], who used the PLSR model to predict the tallgrass prairie leaf pigment and nutritional status with the lowest RMSE of prediction. A reasonable selection of modeling and validation datasets is important to improve the prediction accuracy of the PLSR models. The spectral modeling of leaf nutrients is complex and depends on spectral features [49]. The PSLR has the capability of building linkages between the high dimensional spectral features and the vegetation properties. Reliability of PLSR is due to its ability to address the property of collinearity and over-fitting in the hyperspectral data than other multivariate models [97,98], and hence the PLSR is widely preferred in the hyperspectral analysis [99–101]. The outcome could be due to the low nutrient ranges and weak relationships between nutrients and reflectance that hinder the model development. The results of better prediction of N% with the SVR are consistent with those reported by [38] in the pear ($R^2 = 0.66$) and apple ($R^2 = 0.77$). In [51], satisfactory results for all macronutrients ($R^2 = 0.69$–0.92, RPD = 1.62–3.62) were also observed, with N predicted best followed by P, K, and S. The micronutrients group showed lower prediction accuracy ($R^2$ from 0.19 to 0.86, RPD from 1.09 to 2.69), however, indicated Cu and Zn were best predicted, followed by Fe and Mn. In the current study, we employed PLSR, but other multivariate modeling techniques such as random forest [102], and artificial neural networks [103] can also be used. The constraint of using advanced modeling tools could be that these are pure data-driven approaches, and it might be difficult to interpret the biological processes and significance. Owing to the poor performance of the single PLSR and other multivariate models, a new approach of combining the PLSR with machine learning [20] was attempted to retrieve the leaf nutrient content to improve the accuracy. In general, prediction accuracy improved significantly with the PLSR-combined machine learning models over the single PLSR model. The ELNET, SVR, GPR, MARS, RF, KNN, XGB, NNET, and Cubist combined with the PLSR retrieved N, P, K, Ca, Mg, S, Fe, Mn, Zn, Cu, and B with very good to excellent prediction accuracy with few exceptions such as poor prediction of MARS for K and Zn and KNN for Mg and B, which yielded non-reliable predictions.

The current study explored an approach of using the PLSR in combination with machine learning models of the spectral data as an attempt to retrieve leaf nutrient content, and very few studies have been conducted in this context so far. The use of such an

approach of latent variable modeling reduces the redundancy and dimensionality in the data and speed of computation with a meager loss of information from the original data [104]. Furthermore, the use of visible-NIR (350–1050 nm) spectral data in the study also provides a greater opportunity to upscale it to the field level [20].

An approach identified in the current study would help in offering the guidelines for precision nutrient management in mango crops, which might further help to improve the fruit yield and quality. The present approach is suitable for rapid and reliable estimation of the leaf nutrients at the laboratory level, however, field investigations are needed to upscale this research at the canopy level using the ground-based or airborne hyperspectral remote sensing. A major limitation or constraint to upscale the research at field or canopy level is cloud cover, which coincides with sampling time, i.e., the post-fruiting season in the study region.

## 5. Conclusions

In the current study, spectroscopy-based novel spectral indices, chemometric modeling methods—solo PLSR, PCR, and SVR—and PLSR-based machine learning models were evaluated to predict the mango leaf macro- and micronutrient contents. The approach of spectral indices and chemometrics modeling methods both were inefficient and could not retrieve any of the nutrients satisfactorily. In the study, a combination of linear and non-linear machine learning methods yielded the best predictions. The PLSR-combined machine learning models of the Cubist, SVR, and ELNET were found to be the most robust in predicting most of the nutrients and provided very good to excellent prediction accuracy. The results of the study revealed that the hyperspectral sensing data could be employed to retrieve the foliar nutritional status of the mango. The presented approach is suitable for rapid and reliable estimation of the leaf nutrients at the laboratory level, however, field investigations are needed to upscale this research at the canopy level using ground-based or airborne hyperspectral remote sensing.

**Author Contributions:** Conceptualization, G.R.M. and B.D.; methodology, G.R.M. and B.D.; software, G.R.M., B.D., and R.N.S.; validation, G.R.M. and B.D.; formal analysis, G.R.M., K.P., D.M., A.D., S.M., and R.M.K.; investigation, G.R.M., B.D., I.H., and K.B.; resources, G.R.M. and B.D.; data curation, G.R.M. and B.D.; writing—original draft preparation, G.R.M., B.D., D.M., I.H., and K.B.; investigation, G.R.M., B.D., and K.B.; resources, G.R.M. and B.D.; data curation, G.R.M. and B.D.; writing—review and editing, G.R.M., B.D., D.M., I.H., and K.B.; visualization, G.R.M. and B.D.; supervision, G.R.M. and B.D.; project administration, G.R.M.; funding acquisition, G.R.M.; All authors have read and agreed to the published version of the manuscript.

**Funding:** This research was funded by Science and Engineering Research Board, Department of Science and Technology, Ministry of Science and Technology, Government of India, Grant Number ECR/2017/000282 under the scheme Early Career Research Award. Katja Berger is funded within the EnMAP scientific preparation program under the DLR Space Administration with resources from the German Federal Ministry of Economic Affairs and Energy, grant number 50EE1923.

**Institutional Review Board Statement:** Not applicable.

**Informed Consent Statement:** Not applicable

**Data Availability Statement:** Not applicable

**Acknowledgments:** The authors are thankful to the Indian Council of Agricultural Research, New Delhi, India, and Director of Indian Council of Agricultural Research (ICAR)–Central Coastal Agricultural Research Institute, Old Goa, Goa 403402, India for providing support for this research.

**Conflicts of Interest:** The authors declare no conflict of interest. The funders had no role in the design of the study; in the collection, analyses, or interpretation of data; in the writing of the manuscript, or in the decision to publish the results.

## Appendix A

**Table A1.** Performance of the PLSR-combined machine learning models for calibration and validation to predict the nutrients content mango leaf.

| | Calibration | | | | | | Validation | | | | | | |
| | $R^2_c$ | dindex$_c$ | MBE$_c$ | RMSE$_c$ | RPD$_c$ | RPIQ$_c$ | $R^2_v$ | dindex$_v$ | MBE$_v$ | RMSE$_v$ | RPD$_v$ | RPIQ$_v$ | Summed rank |
|---|---|---|---|---|---|---|---|---|---|---|---|---|---|
| N | | | | | | | | | | | | | |
| ELNET | 0.95 | 0.99 | −0.0002 | 0.028 | 4.50 | 7.96 | 0.94 | 0.99 | −0.0019 | 0.029 | 4.19 | 5.92 | 38 |
| SVR | 0.98 | 0.99 | −0.0007 | 0.019 | 6.48 | 11.46 | 0.85 | 0.95 | −0.0042 | 0.048 | 2.48 | 3.51 | 37 |
| GPR | 0.96 | 0.97 | −0.0017 | 0.035 | 3.53 | 6.25 | 0.84 | 0.92 | −0.0042 | 0.057 | 2.12 | 3.00 | 67 |
| MARS | 0.80 | 0.94 | −0.0005 | 0.056 | 2.22 | 3.92 | 0.58 | 0.86 | 0.0068 | 0.097 | 1.24 | 1.75 | 93 |
| RF | 0.98 | 0.97 | −0.0011 | 0.035 | 3.51 | 6.21 | 0.72 | 0.75 | 0.0006 | 0.082 | 1.47 | 2.07 | 77 |
| KNN | 0.78 | 0.89 | 0.0007 | 0.066 | 1.89 | 3.35 | 0.55 | 0.81 | −0.0103 | 0.082 | 1.47 | 2.08 | 96 |
| XGB | 0.97 | 0.99 | 0.0012 | 0.023 | 5.36 | 9.48 | 0.77 | 0.93 | 0.0001 | 0.057 | 2.11 | 2.98 | 48 |
| NNET | 0.95 | 0.99 | 0.0085 | 0.028 | 4.38 | 7.75 | 0.95 | 0.99 | 0.0066 | 0.029 | 4.22 | 5.96 | 48 |
| Cubist | 0.95 | 0.99 | 0.0003 | 0.028 | 4.42 | 7.83 | 0.94 | 0.99 | −0.0010 | 0.028 | 4.27 | 6.03 | 36 |
| P | | | | | | | | | | | | | |
| ELNET | 0.94 | 0.99 | 0.0001 | 0.007 | 4.25 | 5.59 | 0.92 | 0.98 | −0.0003 | 0.008 | 3.47 | 3.90 | 28 |
| SVR | 0.97 | 0.99 | −0.0002 | 0.006 | 5.39 | 7.10 | 0.89 | 0.97 | −0.0001 | 0.009 | 3.04 | 3.42 | 35 |
| GPR | 0.95 | 0.97 | 0.0000 | 0.009 | 3.53 | 4.64 | 0.89 | 0.96 | −0.0001 | 0.009 | 2.81 | 3.16 | 54 |
| MARS | 0.76 | 0.93 | 0.0003 | 0.015 | 2.06 | 2.71 | 0.58 | 0.87 | −0.0008 | 0.018 | 1.51 | 1.70 | 89 |
| RF | 0.98 | 0.97 | 0.0003 | 0.009 | 3.34 | 4.40 | 0.79 | 0.79 | 0.0009 | 0.017 | 1.57 | 1.77 | 73 |
| KNN | 0.79 | 0.90 | −0.0010 | 0.016 | 1.95 | 2.57 | 0.65 | 0.85 | 0.0012 | 0.016 | 1.63 | 1.84 | 91 |
| XGB | 0.94 | 0.99 | 0.0001 | 0.007 | 4.18 | 5.50 | 0.77 | 0.93 | 0.0003 | 0.013 | 2.10 | 2.37 | 56 |
| NNET | 0.94 | 0.98 | 0.0007 | 0.007 | 4.14 | 5.45 | 0.91 | 0.98 | 0.0004 | 0.008 | 3.34 | 3.77 | 44 |
| Cubist | 0.98 | 0.99 | −0.0002 | 0.004 | 6.92 | 9.10 | 0.91 | 0.98 | −0.0006 | 0.008 | 3.30 | 3.71 | 24 |
| K | | | | | | | | | | | | | |
| ELNET | 0.98 | 0.99 | −0.0001 | 0.043 | 6.38 | 7.96 | 0.97 | 0.99 | 0.0071 | 0.046 | 6.04 | 8.03 | 25 |
| SVR | 0.99 | 1.00 | −0.0023 | 0.035 | 7.69 | 9.60 | 0.92 | 0.97 | 0.0120 | 0.087 | 3.21 | 4.27 | 44 |
| GPR | 0.98 | 0.98 | −0.0029 | 0.068 | 3.98 | 4.97 | 0.93 | 0.95 | 0.0107 | 0.105 | 2.68 | 3.56 | 65 |
| MARS | 0.83 | 0.95 | −0.0004 | 0.112 | 2.43 | 3.04 | 0.76 | 0.92 | 0.0057 | 0.138 | 2.03 | 2.70 | 79 |
| RF | 0.98 | 0.98 | −0.0022 | 0.074 | 3.70 | 4.61 | 0.78 | 0.80 | 0.0183 | 0.179 | 1.57 | 2.08 | 85 |
| KNN | 0.83 | 0.92 | −0.0087 | 0.126 | 2.16 | 2.69 | 0.62 | 0.82 | 0.0170 | 0.180 | 1.56 | 2.07 | 105 |
| XGB | 0.98 | 0.99 | 0.0012 | 0.043 | 6.34 | 7.91 | 0.82 | 0.95 | 0.0103 | 0.118 | 2.38 | 3.16 | 62 |
| NNET | 0.98 | 0.99 | −0.0105 | 0.045 | 6.01 | 7.50 | 0.97 | 0.99 | −0.0071 | 0.052 | 5.38 | 7.14 | 54 |
| Cubist | 0.99 | 1.00 | −0.0007 | 0.025 | 11.06 | 13.80 | 0.97 | 0.99 | 0.0071 | 0.049 | 5.76 | 7.65 | 21 |
| Ca | | | | | | | | | | | | | |
| ELNET | 0.96 | 0.99 | −0.0107 | 0.322 | 4.70 | 5.58 | 0.94 | 0.98 | −0.0535 | 0.318 | 4.08 | 4.94 | 34 |
| SVR | 0.98 | 0.99 | −0.0214 | 0.238 | 6.36 | 7.54 | 0.88 | 0.96 | 0.0001 | 0.475 | 2.73 | 3.31 | 30 |
| GPR | 0.97 | 0.98 | −0.0683 | 0.422 | 3.59 | 4.25 | 0.88 | 0.93 | −0.0285 | 0.552 | 2.35 | 2.85 | 70 |
| MARS | 0.84 | 0.95 | −0.0337 | 0.609 | 2.49 | 2.95 | 0.66 | 0.90 | 0.0129 | 0.759 | 1.71 | 2.07 | 84 |
| RF | 0.98 | 0.98 | −0.0721 | 0.393 | 3.85 | 4.57 | 0.63 | 0.80 | −0.0814 | 0.852 | 1.52 | 1.84 | 83 |
| KNN | 0.83 | 0.87 | −0.1389 | 0.833 | 1.82 | 2.15 | 0.65 | 0.79 | −0.1060 | 0.867 | 1.50 | 1.81 | 107 |
| XGB | 0.97 | 0.99 | −0.0158 | 0.252 | 6.02 | 7.13 | 0.77 | 0.93 | −0.0712 | 0.624 | 2.08 | 2.52 | 49 |
| NNET | 0.97 | 0.99 | −0.0195 | 0.266 | 5.70 | 6.76 | 0.91 | 0.98 | -0.0125 | 0.382 | 3.39 | 4.11 | 36 |
| Cubist | 0.96 | 0.99 | −0.0273 | 0.324 | 4.68 | 5.55 | 0.94 | 0.98 | −0.0701 | 0.320 | 4.05 | 4.90 | 47 |
| Mg | | | | | | | | | | | | | |
| ELNET | 0.95 | 0.99 | 0.0000 | 190.287 | 4.70 | 6.20 | 0.95 | 0.99 | 28.6533 | 219.943 | 4.47 | 6.11 | 28 |
| SVR | 0.98 | 0.99 | −2.5794 | 133.784 | 6.69 | 8.82 | 0.90 | 0.96 | −19.6623 | 352.997 | 2.79 | 3.81 | 27 |
| GPR | 0.97 | 0.98 | 6.0932 | 226.551 | 3.95 | 5.21 | 0.91 | 0.94 | −2.3241 | 393.767 | 2.50 | 3.41 | 54 |
| MARS | 0.78 | 0.94 | 0.0000 | 414.932 | 2.16 | 2.84 | 0.67 | 0.90 | −39.8115 | 565.089 | 1.74 | 2.38 | 78 |
| RF | 0.98 | 0.98 | 2.9559 | 227.748 | 3.93 | 5.18 | 0.71 | 0.82 | −50.5409 | 618.348 | 1.59 | 2.17 | 77 |
| KNN | 0.80 | 0.90 | 82.2845 | 455.592 | 1.96 | 2.59 | 0.80 | 0.88 | 44.4215 | 528.350 | 1.86 | 2.54 | 84 |
| XGB | 0.97 | 0.99 | -3.8362 | 144.384 | 6.20 | 8.17 | 0.86 | 0.95 | 11.3587 | 381.799 | 2.58 | 3.52 | 39 |
| NNET | 0.04 | 0.45 | −1163.8440 | 1640.410 | 0.55 | 0.72 | 0.07 | 0.47 | -1340.4810 | 1727.948 | 0.57 | 0.78 | 108 |
| Cubist | 0.95 | 0.99 | 19.8129 | 191.229 | 4.68 | 6.17 | 0.95 | 0.99 | 49.4772 | 222.529 | 4.42 | 6.04 | 45 |
| S | | | | | | | | | | | | | |
| ELNET | 0.96 | 0.99 | −0.0001 | 0.009 | 4.95 | 6.55 | 0.95 | 0.99 | −0.0015 | 0.011 | 4.48 | 6.80 | 26 |
| SVR | 0.98 | 0.99 | −0.0002 | 0.007 | 6.18 | 8.18 | 0.87 | 0.96 | −0.0016 | 0.018 | 2.69 | 4.09 | 40 |
| GPR | 0.97 | 0.98 | −0.0006 | 0.013 | 3.62 | 4.79 | 0.89 | 0.93 | −0.0025 | 0.021 | 2.36 | 3.57 | 68 |
| MARS | 0.78 | 0.93 | −0.0006 | 0.021 | 2.13 | 2.82 | 0.62 | 0.88 | −0.0015 | 0.031 | 1.61 | 2.44 | 85 |
| RF | 0.98 | 0.97 | −0.0012 | 0.013 | 3.46 | 4.57 | 0.78 | 0.76 | −0.0069 | 0.033 | 1.48 | 2.25 | 84 |
| KNN | 0.76 | 0.85 | −0.0026 | 0.027 | 1.70 | 2.26 | 0.70 | 0.77 | −0.0083 | 0.034 | 1.46 | 2.22 | 105 |
| XGB | 0.98 | 0.99 | −0.0001 | 0.007 | 6.97 | 9.22 | 0.84 | 0.94 | −0.0015 | 0.020 | 2.42 | 3.68 | 37 |
| NNET | 0.96 | 0.98 | 0.0072 | 0.012 | 3.87 | 5.12 | 0.95 | 0.98 | 0.0057 | 0.012 | 3.99 | 6.05 | 56 |
| Cubist | 0.96 | 0.99 | 0.0003 | 0.009 | 4.86 | 6.43 | 0.95 | 0.99 | −0.0008 | 0.011 | 4.38 | 6.64 | 39 |
| Fe | | | | | | | | | | | | | |
| ELNET | 0.96 | 0.99 | 0.0000 | 5.714 | 4.95 | 7.13 | 0.94 | 0.98 | 0.4834 | 6.649 | 4.19 | 5.10 | 33 |
| SVR | 0.98 | 0.99 | −0.2295 | 4.427 | 6.38 | 9.20 | 0.91 | 0.97 | −0.2120 | 8.560 | 3.26 | 3.96 | 31 |
| GPR | 0.96 | 0.98 | 0.0700 | 7.743 | 3.65 | 5.26 | 0.91 | 0.95 | −0.1209 | 10.470 | 2.66 | 3.24 | 54 |
| MARS | 0.77 | 0.93 | 0.0000 | 13.670 | 2.07 | 2.98 | 0.68 | 0.90 | −0.5475 | 15.952 | 1.75 | 2.13 | 71 |
| RF | 0.98 | 0.98 | −0.1374 | 7.347 | 3.85 | 5.55 | 0.68 | 0.82 | −1.5566 | 17.722 | 1.57 | 1.91 | 70 |
| KNN | 0.76 | 0.89 | −0.1584 | 15.173 | 1.86 | 2.69 | 0.60 | 0.81 | −1.2094 | 18.478 | 1.51 | 1.83 | 93 |
| XGB | 0.97 | 0.99 | 0.1141 | 4.796 | 5.89 | 8.50 | 0.85 | 0.96 | −1.2427 | 10.731 | 2.60 | 3.16 | 47 |
| NNET | 0.22 | 0.68 | −4.9762 | 34.855 | 0.81 | 1.17 | 0.24 | 0.68 | −7.4574 | 33.930 | 0.82 | 1.00 | 108 |
| Cubist | 0.96 | 0.99 | −0.0352 | 5.720 | 4.94 | 7.12 | 0.94 | 0.99 | 0.4824 | 6.580 | 4.23 | 5.15 | 33 |

**Table A1.** *Cont.*

| | Calibration | | | | | | Validation | | | | | | |
|---|---|---|---|---|---|---|---|---|---|---|---|---|---|
| | $R^2_c$ | dindex$_c$ | MBE$_c$ | RMSE$_c$ | RPD$_c$ | RPIQ$_c$ | $R^2_v$ | dindex$_v$ | MBE$_v$ | RMSE$_v$ | RPD$_v$ | RPIQ$_v$ | Summed rank |
| **Mn** | | | | | | | | | | | | | |
| ELNET | 0.75 | 0.88 | -0.5831 | 205.813 | 1.19 | 1.29 | 0.88 | 0.96 | −23.2350 | 70.000 | 2.74 | 3.31 | 50 |
| SVR | 0.94 | 0.97 | −10.7918 | 71.627 | 3.41 | 3.70 | 0.80 | 0.93 | −14.1564 | 89.239 | 2.15 | 2.59 | 30 |
| GPR | 0.93 | 0.94 | −28.2337 | 98.969 | 2.46 | 2.68 | 0.79 | 0.89 | −28.6016 | 103.478 | 1.86 | 2.24 | 58 |
| MARS | 0.77 | 0.93 | −17.9819 | 118.470 | 2.06 | 2.24 | 0.46 | 0.78 | −18.7017 | 141.898 | 1.35 | 1.63 | 70 |
| RF | 0.97 | 0.95 | −28.0254 | 94.096 | 2.59 | 2.81 | 0.45 | 0.67 | −30.0085 | 151.782 | 1.27 | 1.53 | 67 |
| KNN | 0.78 | 0.80 | −50.8969 | 159.806 | 1.53 | 1.66 | 0.39 | 0.66 | −44.4179 | 158.226 | 1.21 | 1.46 | 98 |
| XGB | 0.94 | 0.98 | −3.8812 | 68.932 | 3.54 | 3.84 | 0.72 | 0.92 | −14.6053 | 102.248 | 1.88 | 2.26 | 33 |
| NNET | 0.61 | 0.87 | 40.2527 | 158.673 | 1.54 | 1.67 | 0.62 | 0.88 | 26.0067 | 133.700 | 1.44 | 1.73 | 78 |
| Cubist | 0.75 | 0.85 | 17.9304 | 259.424 | 0.94 | 1.02 | 0.87 | 0.97 | −13.9173 | 71.479 | 2.69 | 3.24 | 56 |
| **Zn** | | | | | | | | | | | | | |
| ELNET | 0.94 | 0.98 | −0.1776 | 4.547 | 3.82 | 4.78 | 0.95 | 0.99 | −0.0372 | 0.701 | 4.71 | 7.48 | 39 |
| SVR | 0.98 | 0.99 | −0.1855 | 2.407 | 7.22 | 9.02 | 0.93 | 0.97 | 0.0334 | 1.034 | 3.19 | 5.07 | 30 |
| GPR | 0.97 | 0.98 | −1.1132 | 4.654 | 3.73 | 4.67 | 0.93 | 0.94 | −0.0425 | 1.329 | 2.48 | 3.95 | 67 |
| MARS | 0.77 | 0.93 | −0.5952 | 8.370 | 2.08 | 2.59 | 0.64 | 0.89 | 0.1141 | 1.970 | 1.67 | 2.66 | 90 |
| RF | 0.98 | 0.97 | −1.1889 | 4.972 | 3.49 | 4.37 | 0.75 | 0.67 | −0.0365 | 2.436 | 1.35 | 2.15 | 84 |
| KNN | 0.82 | 0.88 | −2.0310 | 9.361 | 1.86 | 2.32 | 0.69 | 0.79 | −0.2792 | 2.157 | 1.53 | 2.43 | 101 |
| XGB | 0.96 | 0.99 | −0.2439 | 3.266 | 5.32 | 6.65 | 0.82 | 0.95 | 0.1252 | 1.396 | 2.36 | 3.76 | 59 |
| NNET | 0.97 | 0.98 | 2.9880 | 4.590 | 3.78 | 4.73 | 0.95 | 0.99 | −0.0083 | 0.728 | 4.53 | 7.20 | 48 |
| Cubist | 0.98 | 0.99 | 0.1900 | 2.887 | 6.02 | 7.52 | 0.95 | 0.99 | −0.0161 | 0.700 | 4.71 | 7.49 | 22 |
| **Cu** | | | | | | | | | | | | | |
| ELNET | 0.88 | 0.96 | −0.0065 | 0.204 | 2.26 | 3.63 | 0.90 | 0.96 | 0.0101 | 0.204 | 2.41 | 3.06 | 50 |
| SVR | 0.98 | 0.99 | −0.0162 | 0.070 | 6.58 | 10.57 | 0.90 | 0.97 | −0.0220 | 0.163 | 3.01 | 3.82 | 17 |
| GPR | 0.95 | 0.97 | −0.0713 | 0.147 | 3.13 | 5.02 | 0.90 | 0.93 | −0.0769 | 0.211 | 2.33 | 2.96 | 52 |
| MARS | 0.69 | 0.91 | −0.0371 | 0.276 | 1.67 | 2.69 | 0.74 | 0.92 | −0.0647 | 0.259 | 1.90 | 2.42 | 81 |
| RF | 0.97 | 0.96 | −0.0772 | 0.164 | 2.80 | 4.50 | 0.64 | 0.73 | -0.1163 | 0.361 | 1.36 | 1.73 | 84 |
| KNN | 0.74 | 0.83 | −0.1726 | 0.311 | 1.48 | 2.38 | 0.73 | 0.82 | −0.1652 | 0.333 | 1.48 | 1.88 | 101 |
| XGB | 0.96 | 0.99 | −0.0055 | 0.100 | 4.63 | 7.43 | 0.71 | 0.92 | −0.0281 | 0.274 | 1.79 | 2.28 | 57 |
| NNET | 0.92 | 0.95 | −0.0777 | 0.173 | 2.66 | 4.27 | 0.87 | 0.93 | −0.0861 | 0.222 | 2.22 | 2.82 | 71 |
| Cubist | 0.98 | 0.99 | −0.0103 | 0.079 | 5.85 | 9.38 | 0.88 | 0.97 | −0.0156 | 0.191 | 2.57 | 3.27 | 27 |
| **B** | | | | | | | | | | | | | |
| ELNET | 0.94 | 0.98 | −0.1776 | 4.547 | 3.82 | 4.78 | 0.93 | 0.98 | −0.8644 | 4.294 | 3.64 | 5.64 | 35 |
| SVR | 0.98 | 0.99 | −0.1855 | 2.407 | 7.22 | 9.02 | 0.92 | 0.98 | 0.5814 | 4.542 | 3.44 | 5.33 | 22 |
| GPR | 0.97 | 0.98 | −1.1132 | 4.654 | 3.73 | 4.67 | 0.91 | 0.96 | −0.1763 | 5.301 | 2.94 | 4.57 | 58 |
| MARS | 0.77 | 0.93 | −0.5952 | 8.370 | 2.08 | 2.59 | 0.51 | 0.83 | −1.9934 | 11.137 | 1.40 | 2.18 | 97 |
| RF | 0.98 | 0.97 | −1.1889 | 4.972 | 3.49 | 4.37 | 0.56 | 0.77 | −1.0856 | 10.938 | 1.43 | 2.22 | 83 |
| KNN | 0.82 | 0.88 | −2.0310 | 9.361 | 1.86 | 2.32 | 0.71 | 0.88 | −1.5863 | 8.885 | 1.76 | 2.73 | 93 |
| XGB | 0.96 | 0.99 | −0.2439 | 3.266 | 5.32 | 6.65 | 0.77 | 0.92 | −2.1715 | 7.787 | 2.00 | 3.11 | 60 |
| NNET | 0.97 | 0.98 | 2.9880 | 4.590 | 3.78 | 4.73 | 0.94 | 0.97 | 2.9940 | 6.095 | 2.56 | 3.98 | 61 |
| Cubist | 0.98 | 0.99 | 0.1900 | 2.887 | 6.02 | 7.52 | 0.92 | 0.98 | −0.5982 | 4.577 | 3.41 | 5.29 | 31 |

N, nitrogen; P, phosphorus; K, potassium; Ca, calcium; Mg, magnesium; S, sulfur; Fe, iron; Mn, manganese; Zn, zinc; Cu, copper; B, boron; $R^2$, regression coefficient; MBE, mean bias error; RMSE, root means square error; RPD, ratio of performance to deviation; RPIQ, ratio of performance to inter-quartile distance; ELNET, elastic net; SVR, support vector regression; GPR, Gaussian process regression; MARS, multivariate adaptive regression splines; RF, random forest; KNN, k-nearest neighbors; XGB, extreme gradient boosting; NNET, neural network.

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
