# Peer review of "Monitoring the Foliar Nutrients Status of Mango Using Spectroscopy-Based Spectral Indices and PLSR-Combined Machine Learning Models"

_remotesensing, doi:10.3390/rs13040641_

Round 1

Reviewer 1 Report

This study focus to use remote sensing to characterize the foliar nutrient status of mango through the spectral indices, multivariate analysis, chemometrics and machine learning modeling of the spectral data. The PLSR-combined machine learning models of the cubist, SVR and ELNET were found to be the most robust in predicting most of the nutrients and provided very good to excellent prediction accuracy. The results of the study revealed that the hyperspectral sensing data could be employed to retrieve the foliar nutritional status of the mango.

This research falls well within the scope of this journal. The paper is well structured and well-written, the main part of the methods seems to make sense. I have some concerns about the introduction, method and results presentations. I think it needs some major modifications before it can be accepted for publication in this journal.

General comments:

  1. I suggest the authors consider whether to keep the word ‘novel’ in the title of the paper. The normalized difference vegetation indices and ration vegetation indices, as well as band-band contour plotting methods are widely used in previous studies.

2.The introduction section should be reorganized. The content about statistical approaches and analysis models could be combined as one part.

  1. The introduction of summed rank evaluating algorithm for different models should be added in section 2.5.

4.The latent variables (LVs) for different leaf nutrient content which generated from the PLSR model should be listed and analyzed in section 3.2.

  1. The format of the references needs to be modified according to the journal's requirements.

Author Response

This study focus to use remote sensing to characterize the foliar nutrient status of mango through the spectral indices, multivariate analysis, chemometrics and machine learning modeling of the spectral data. The PLSR-combined machine learning models of the cubist, SVR and ELNET were found to be the most robust in predicting most of the nutrients and provided very good to excellent prediction accuracy. The results of the study revealed that the hyperspectral sensing data could be employed to retrieve the foliar nutritional status of the mango.

This research falls well within the scope of this journal. The paper is well structured and well-written, the main part of the methods seems to make sense. I have some concerns about the introduction, method and results presentations. I think it needs some major modifications before it can be accepted for publication in this journal.

Response: We acknowledge the comment for major modifications. We have studied the comments thoroughly and responded as a point-by-point reply.

General comments:

Point 1: I suggest the authors consider whether to keep the word ‘novel’ in the title of the paper. The normalized difference vegetation indices and ration vegetation indices, as well as band-band contour plotting methods are widely used in previous studies.

Response: The word ‘novel’ has been removed from the title.

Point 2: The introduction section should be reorganized. The content about statistical approaches and analysis models could be combined as one part.

Response: The paragraph number 2, 3 and 4 on statistical approaches and analyses models of the section ‘1. Introduction’ has been combined as it covers information about the statistical approaches and analysis models.

Point 3: The introduction of summed rank evaluating algorithm for different models should be added in section 2.5.

Response: The introduction about the summed rank evaluating algorithms for different models have been revised and added to the section 2.5 as below,

It is difficult to decide the best performing model evaluation parameters like R2, RMSE, RPD, RPIQ, etc. individually. So, a composite summed rank based on these parameters was developed considering the performance of each model parameter wise for the calibration and validation. Ranking of each model evaluation parameter for a particular nutrient in calibration or validation was done using the RANK.AVG function of the Microsoft Excel. The ranks of calibration and validation were summed separately and altogether and it was referred as a summed rank for a particular model. The model with the least rank predicted the nutrient with the greatest accuracy and the one with the highest rank had the poorest prediction accuracy.

Point 4: The latent variables (LVs) for different leaf nutrient content which generated from the PLSR model should be listed and analyzed in section 3.2.

Response: The latent variables (LVs) for different leaf nutrient content generated from PLSR are listed in the revised manuscript in the section as below,

The optimum number of latent variables (LVs) were generated and selected using PLSR and 10 fold cross-validation and used as predictor variables for machine learning model development. The LVs are linear combinations of all the input variables but orthogonal to each other which helps to reduce collinearity. The number of the LVs for N, P, K, Ca, Mg, S, Fe, Mn, Zn, Cu and B selected were 1, 6, 5, 5, 7, 10, 5, 5, 5, 1 and 6, respectively.

Point 5: The format of the references needs to be modified according to the journal's requirements.

Response: The format of the references has been modified as per the journal’s requirements.

Reviewer 2 Report

Although the topic of the manuscript is of wide interest in remote sensing community, there are some issues the author may need to address in the revision.

2.2. Spectral measurements
You should have offered some details of Savitzky-Golay filtering (the window length and the filter order).

Did you use leaf clip? If no, reflectance is also affected by BRF and diffuse light and then it is difficult to evaluate the validity of the obtained reflectance data since the explanations about settings and environment were not enough.

2.5. Development of nonparametric regression models
Which hyperparameters did you optimize?

Did you conduct the separating procedure only once? for more robust conclusions, some repetitions were required.

4.3. Chemometrics and machine learning regression modeling
Which wavelengths were effective for estimation?

Author Response

Comments and Suggestions for Authors

Although the topic of the manuscript is of wide interest in remote sensing community, there are some issues the author may need to address in the revision.

Response: We acknowledge the comments for a revision of the manuscript. We have studied the comments thoroughly and responded as point-by-point reply.

Point 1: 2.2. Spectral measurements
You should have offered some details of Savitzky-Golay filtering (the window length and the filter order).

Response: The details of the Savitzky-Golay filtering has been added as below,

The spectral data were further smoothed using Savitzky–Golay filtering across a fifteen bands moving window (a window length of 15 and zero polynomial order) to reduce the noise using “prospectr” package in R software version 3.5.2 (Stevens et al. 2013).

Point 2: Did you use leaf clip? If no, reflectance is also affected by BRF and diffuse light and then it is difficult to evaluate the validity of the obtained reflectance data since the explanations about settings and environment were not enough.

Response: The spectral observations of adaxial surface of mango leaves were taken within a black box to reduce the impact of stray light (Picture given below). It was ensured that the leaves cover the full field of view of the foreoptic (pistol grip). The spectral observations were taken at nadir position to reduce the impact due bidirectional reflectance. For each leaf sample, an average of five measurements was considered as a representative spectral signature.

Picture: Set up for the spectral measurements

The details in this regards have been added to the section ‘2.2 Spectral Measurements’ of the ‘2. Materials and Methods’ Section.

Point 3: 2.5. Development of nonparametric regression models
Which hyperparameters did you optimize?

Response: Different hyperparameters optimized for each of the machine learning models used in the study were mentioned in the revised manuscript in the section 2.5 Development of non-parametric regression models, as below

The hyperparameters which were optimized for each machine learning model were as ELNET - alpha, lambda; SVR - sigma, C; GPR – sigma; MARS - nprune, degree; RF – mtry; splitrule, min.node.size; KNN - Number of neighbors (k); XGB - nrounds, max_depth, eta, gamma, colsample_bytree, min_child_weight, subsample; NNET- size, decay and Cubist -committees, neighbors.

Point 4: Did you conduct the separating procedure only once? for more robust conclusions, some repetitions were required.

Response: We have divided the whole dataset randomly into two parts using ‘sample’ function of R software. Out of the total dataset, 70% was used for calibration and remaining 30% was used for validation. Then every machine learning model was calibrated using training dataset using 10 fold cross-validation with 5 repetitions and thus each model was run 50 times (10*5 = 50).

Point 5: 4.3. Chemometrics and machine learning regression modeling
Which wavelengths were effective for estimation?

Response: We have used latent variables (LVs) generated from the PLSR model as input variables for developing different linear and non-linear models. The LVs are linear combinations of all the input variables (wavelengths) but orthogonal to each other which helps to reduce collinearity. So, it is not possible to identify effective wavelengths from machine learning regression models. However, we have identified wavelengths effective for estimation of mango leaf nutrients using contour plotting approach and the performance of the vegetation indices using the effective wavelengths has been studied (Section 3.2, Table 3, Figure 4 and 5)

Round 2

Reviewer 1 Report

The manuscript has been improved by the authors. The scientific soundness have been increased significantly. Results and Discussion are in much better shape and  the results are presented in a more efficient and effective way.  

Reviewer 2 Report

The authors added some elements to enrich manuscript and I think this paper can now be accepted for publication.